



# Heterogeneous reactions of NO$_2$ with CaCO$_3$-(NH$_4$)$_2$SO$_4$ mixtures at different relative humidities

**Fang Tan, Shengrui Tong, Bo Jing, Siqi Hou, Qifan Liu, Kun Li, Ying Zhang, and Maofa Ge**

Beijing National Laboratory for Molecular Sciences (BNLMS), State Key Laboratory for Structural Chemistry of Unstable and Stable Species, Institute of Chemistry, Chinese Academy of Sciences, 100190, Beijing, China

5  Correspondence to Maofa Ge (gemaofa@iccas.ac.cn) and Shengrui Tong (tongsr@iccas.ac.cn)



## Abstract

In this work, the heterogeneous reactions of $NO_2$ with $CaCO_3$-$(NH_4)_2SO_4$ mixtures with a series of weight percentage (wt%) of $(NH_4)_2SO_4$ were investigated using a diffuse reflectance infrared Fourier transform spectroscopy (DRIFTS) at different relative humidity (RH) values.

For comparison, the heterogeneous reactions of $NO_2$ with pure $CaCO_3$ particles and pure $(NH_4)_2SO_4$ particles, as well as the reaction of $CaCO_3$ with $(NH_4)_2SO_4$ particles were also studied. The results indicated that $NO_2$ did not show any significant uptake on $(NH_4)_2SO_4$ particles, and it reacted with $CaCO_3$ particles to form calcium nitrate under both dry and wet conditions. The heterogeneous reactions of $NO_2$ with $CaCO_3$-$(NH_4)_2SO_4$ mixtures were

markedly dependent on RH. Calcium nitrate was formed from the heterogeneous reactions at all the RHs investigated, whereas $CaSO_4\ 0.5H_2O$ (bassanite), $CaSO_4\ 2H_2O$ (gypsum) and $(NH_4)_2Ca(SO_4)_2\ H_2O$ (koktaite) were produced depending on RH. Under dry condition, the $NO_3^-$ mass concentrations for the $CaCO_3$-$(NH_4)_2SO_4$ mixtures had a negative linear relation with the mass fraction of $(NH_4)_2SO_4$ in the mixtures. In this condition, the heterogeneous

uptake of $NO_2$ on the mixtures was similar to that on $CaCO_3$ particles. Under wet conditions, the $CaCO_3$-$(NH_4)_2SO_4$ mixtures exhibited a promotive effect on the heterogeneous uptake of $NO_2$ and the formation of nitrate, especially at medium RHs. In addition, the heterogeneous uptake of $NO_2$ on the mixtures of $CaCO_3$ and $(NH_4)_2SO_4$ was found to favor the formation of bassanite and gypsum due to the decomposition of $CaCO_3$ and the coagulation of $Ca^{2+}$ and

$SO_4^{2-}$. A possible reaction mechanism was proposed and atmospheric implications were discussed.





## 1. Introduction

Haze with high level of fine particulate matter with diameters less than 2.5 μm ($PM_{2.5}$) occurs frequently in China in recent years (Fang et al., 2009; Kulmala, 2015). Emissions of gases pollutants, e.g., $SO_2$, $NO_x$, $NH_3$, and volatile organic compounds (VOCs), result in a series of

atmospheric chemical reactions, which are responsible for the formation of secondary particles and the occurrence of haze (Zhang et al., 2015; Wang et al., 2013; Guo et al., 2014). Chemical analyses show that sulfate, nitrate, and ammonium are the major aerosol constituents of $PM_{2.5}$ (Yang et al., 2011; Huang et al., 2014). Pathakl et al. (2009) discovered that nitrate concentration showed a correlation with sulfate concentration as well as the RH

value in ammonium-poor areas. Kong et al. (2014a) found strong negative correlation between the mass fraction of nitrate and that of sulfate in acidic atmospheric particles during air pollution episodes. Although atmospheric particulate sulfate, nitrate, and ammonium were found to be correlated by numerous field measurements in different locations (Sullivan et al., 2007; Quan et al., 2008; Duan et al., 2003; Possanzini et al., 1999; Querol et al., 1998), there

is still a lack of knowledge to explain the significant relevance.

Mineral dust is a major fraction of airborne particulate matter on a global scale (Tegen et al., 1996) with an estimated annual emission of 1000-3000 Tg of solids into the troposphere (Li et al., 1996). Mineral aerosols provide significant reactants and reactive sites for atmospheric heterogeneous reactions (Usher et al., 2003). Modeling studies indicated that

mineral aerosols was highly associated with nitrate formation in the atmosphere (Dentener et al., 1996). Calcium carbonate represents an important and reactive mineral dust component, approximately accounting for 20-30% of the total dust loading (Usher et al., 2003; Li et al., 2006; Al-Hosney and Grassian, 2005; Prince et al., 2007). Calcium carbonate particle is converted to calcium nitrate after exposing to nitrogen oxides and $HNO_3$ in the atmosphere

(Li et al., 2009; Laskin et al., 2005). Field measurements reveal that mineral dust particles are often mixed with ammonium sulfate aerosols through coagulation during long-range transport (Levin et al., 1996; Zhang et al., 2000). Korhonen et al. (2003) suggested that ammonium sulfate coating of mineral dust by heterogeneous nucleation of $H_2SO_4$, $NH_3$, and $H_2O$ could occur at atmospheric sulphuric acid concentration. Additionally, Mori et al. (1998) have





found the coagulation between $CaCO_3$ and $(NH_4)_2SO_4$ could form koktaite and gypsum, attributing to the interaction of ions under humid condition. Ma et al. (2013) also discovered that mixed $CaCO_3$-$(NH_4)_2SO_4$ particles had synergistic effects on the formation of gypsum in the humidifying-dehumidifying processes.

A few studies have shown that coexisting components play a role in the heterogeneous uptake of trace gases on atmospheric particles. Kong et al. (2014b) found that coexisting nitrate could significantly accelerate the formation rate of sulfate on hematite surface, resulting in surface-adsorbed $HNO_3$, gas-phase $N_2O$ and HONO productions. Zhao et al. (2013) found that coexisting surface nitrate had different effects on the uptake of $H_2O_2$ on

mineral particle surfaces depending on RH. The catalysis and basic coexists could increase the formation of sulfate on NaCl particle surfaces (Li et al., 2007). To the best of our knowledge, the heterogeneous reaction of atmospheric trace gases on mixed $CaCO_3$-$(NH_4)_2SO_4$ particles has not been reported.

    Furthermore, an increase in tropospheric $NO_2$ concentration has been observed in recent

years across many developing regions due to fossil fuel combustion and biomass burning (Zhang et al., 2007; Sheel et al., 2010; Ghude et al., 2009; Shi et al., 2008; Richter et al., 2005; Irie et al., 2005). Atmospheric $NO_2$ concentration ranges from 70 part per billion (ppb) during photochemical smog events to 100-500 ppb in polluted urban environment (Huang et al., 2015; Zamaraev et al., 1994). $NO_2$ is one such critical anthropogenic gaseous pollutant,

which reduces air quality and affects global tropospheric chemistry. $NO_2$ plays a crucial role in the photochemical induced catalytic production of ozone, leading to photochemical smog and increasing tropospheric ozone concentration (Volz and Kley, 1988). Moreover, the heterogeneous reactions of $NO_2$ can also lead to the deposition of nitric acid, as well as the formation of gas phase HONO (Jaegle et al., 1998; Brimblecombe and Stedman, 1982;

Goodman, 1999). Furthermore, the heterogeneous uptake of $NO_2$ on mineral aerosols was responsible for the nitrate accumulation in dust events (Usher et al., 2003). A number of laboratory studies investigated the heterogeneous reaction of $NO_2$ with mineral dust (Underwood et al. 1999b; Borensen et al., 2000; Finlayson-Pitts et al., 2003; Liu et al., 2015; Guan et al., 2014). Miller and Grassian (1998) discovered that $NO_2$ reacted with $Al_2O_3$ and





TiO$_2$ particles to form surface nitrite and nitrate. Underwood et al. (1999a) measured the uptake coefficients of NO$_2$ on Al$_2$O$_3$, TiO$_2$, and Fe$_2$O$_3$ particles using a Knudsen cell. Li et al. (2010) determined the Brunauer-Emmett-Teller (BET) area-corrected initial uptake coefficients to be 10$^{-9}$ and 10$^{-8}$ for the heterogeneous uptake of NO$_2$ on CaCO$_3$ particles under

dry and wet conditions, respectively. However, there are big gaps between the results of modeling studies and field measurements about the quantities and accumulation of nitrate, especially in haze periods (Zheng et al., 2015).

In the present study, the heterogeneous reactions of NO$_2$ with the mixtures of CaCO$_3$ and (NH$_4$)$_2$SO$_4$, pure CaCO$_3$ aprticles, and pure (NH$_4$)$_2$SO$_4$ particles at different RHs were

investigated using a DRIFTS reactor. The surface adsorbed products were monitored and the uptake coefficients of NO$_2$ were determined. The aim of this work is to explore the kinetics and mechanism of the heterogeneous reactions of NO$_2$ with CaCO$_3$-(NH$_4$)$_2$SO$_4$ mixtures and its relevance to RH. The results are helpful for further exploring the correlations among particulate nitrate, sulfate, and ammonium concentration in the atmosphere and partly

contribute to understanding of multicomponent reaction systems in practical environment system.

## 2. Experimental

CaCO$_3$ (99.5 %) and (NH$_4$)$_2$SO$_4$ (99.9%) were purchased from Alfa Aesar. CaCO$_3$ and (NH$_4$)$_2$SO$_4$ were mechanically mixed and grinded together in order to obtain uniform

mixtures with the mass percentage of (NH$_4$)$_2$SO$_4$ in the mixtures ranged from 10% to 93% (wt%) (which were denoted as FAS-10, FAS-20, FAS-40, FAS-57, FAS-75, FAS-87, and FAS-93, respectively). The BET surface areas of pure CaCO$_3$ and (NH$_4$)$_2$SO$_4$ particles were determined to be 8.15 and 0.19 m$^2$ g$^{-1}$, respectively, (Autosorb-1-MP automatic equipment (Quanta Chrome Instrument Co.)). The BET area of the mixtures were determined to be 8.06,

6.62, 4.54, 3.21, 2.34, 1.67, and 0.89 m$^2$ g$^{-1}$ corresponding to the mixtures mentioned above. NO$_2$ (0.1%, Beijing Huayuan Gas Chemical Industry Co., Ltd.) and N$_2$ (>99.999%, Beijing Tailong Electronics Co., Ltd.) were used in this study.

In the gas supply system, N$_2$ was split into two streams; one was dehumidified by silica gel





and molecular sieve to insure RH less than 1% which was called dry condition, the other one was humidified by bubbling through ultrapure water. The flux of dry $N_2$, humid $N_2$, and $NO_2$ were adjusted to reach expected RH (<1%, 40%, 60%, and 85% RH) conditions with the total flow of 400 sccm, using mass flow controllers (Beijing Sevenstar electronics Co., LTD).

Concentration of $NO_2$ entering reactor was diluted to $2.6 \times 10^{15}$ molecules $cm^{-3}$ by mixing with $N_2$. RH and temperature of the inflow of sample cell were measured using a commercial humidity and temperature sensor (HMT330; Vaisala) with a measurement accuracy of $\pm 1\%$ RH and $\pm 0.2^{\circ}C$, respectively.

    In situ DRIFTS experiment was used to monitor reactions in real time without interrupting

the reaction processes and provide mechanistic details and kinetic data (Vogt and Finlaysonpitts, 1994). Infrared spectra of sample surfaces were recorded with a Nicolet FTIR Spectrometer 6700, which was equipped with a liquid-nitrogen-cooled narrow band mercury-cadmium-telluride (MCT) detector and DRIFTS optics (Model CHC-CHA-3, Harrick Scientific Corp.). The DRIFTS equipment has been described elsewhere (Tong et al.,

2010). The spectra were measured at a resolution of 4 $cm^{-1}$ in the spectral range from 4000 to 650 $cm^{-1}$. Each spectrum was generally averaged from 100 scans with a time resolution of 40 s. In situ DRIFTS experiments were carried out on $CaCO_3$-$(NH_4)_2SO_4$ mixtures, $CaCO_3$ particles, and $(NH_4)_2SO_4$ particles, respectively. About 30 mg samples were placed into the stainless steel sample holder (10 mm diameter, 0.5 mm depth). The investigated samples were

exposed to pure nitrogen with expected RH for 20 minutes to establish adsorption equilibrium. Then infrared spectra of the unreacted powder samples were collected as background so that reaction products were observed as positive adsorption bands while losses of surface species as negative adsorption bands. Subsequently, $NO_2$ was introduced into the reaction chamber at a stable RH for 120 min. All the spectra were automatically collected

through a Series program in OMNIC software.

    The products formed on the samples after reaction with $NO_2$ were analyzed by ion chromatography. The filtered solution was analyzed by using a Dionex ICS 900system, equipped with a Dionex AS 14A analytical column and a conductivity detector (DS5). The reacted samples were sonicated for 20 min in 8 ml ultrapure water.



## 3. Results and discussion

### 3.1 Surface products characterization

Figure 1 represents the IR spectra of surface products when the samples were exposed to $NO_2$

for 120 min at different RHs. Under dry condition (Fig. 1a), absorption bands centered at 746, 816, 1040, 1300, and 1330 cm$^{-1}$ which were assigned to surface nitrate could be observed on $CaCO_3$ particle surfaces (FAS-0) and the mixtures (Goodman et al., 2001; Goodman et al., 2000; Al-Hosney and Grassian, 2005). Moreover, peaks at 1630 and 3540 cm$^{-1}$ were assigned to crystal hydrate water in calcium nitrate (Li et al., 2010). It suggested that calcium nitrate

was formed on $CaCO_3$ particle surfaces and the mixtures of $CaCO_3$ and $(NH_4)_2SO_4$. The detailed vibrational assignments were listed in Table 1. Two peaks observed at 1689 and 838 cm$^{-1}$ could be attributed to the $\nu(C=O)$ and $\delta_{oop}(CO_3)$ of adsorbed carbonic acid, respectively, indicating that carbonic acid acted as an intermediate production under dry condition (Al-Hosney, 2004; Al-Abadleh et al., 2004). Besides, adsorbed nitric acid was also formed

with peaks centered at 1710 and 1670 cm$^{-1}$, which were assigned to the asymmetric stretching of adsorbed nitric acid (Goodman, 1999). At the same time, negative bands ranging from 2800 to 3400 cm$^{-1}$ could be ascribed to the loss of surface adsorbed water and negative peaks at 3640 and 3690 cm$^{-1}$ were corresponding to the two types of hydroxyl ions on $CaCO_3$ particle surfaces (Kuriyavar et al., 2000). No obvious negative peaks could be observed when

the samples exposed to dry pure nitrogen for 120 min which indicated that surface adsorbed water and hydroxyl ions participated in the reaction.

Compared with the spectrum of FAS-0, several additional weak absorptions appeared at 1008, 1096, 1155 cm$^{-1}$ on the $CaCO_3$-$(NH_4)_2SO_4$ mixtures, which could be attributed to the vibration modes of $SO_4$ tetrahedra in $CaSO_4$ 0.5$H_2O$ (bassanite) (Prasad, 2005; Liu et al.,

2009). The vibration modes of water group in bassanite were too weak to be observed. In addition, the peak at 1215 cm$^{-1}$ slightly grew in intensity during the whole heterogeneous reaction period of $NO_2$ with the mixtures, whereas it grew fast at the early stage of the reaction of $NO_2$ with $CaCO_3$ particles, and then diminished after reaching a maximum value



at about 30 min (see Fig. S1). This band described before was ascribed to nitrite species, which would convert to nitrate as the reaction proceeded (Miller and Grassian, 1998; G. M. Underwood, 1999b; Wu et al., 2013). To probe this product, samples after reaction with $NO_2$ for different times were detected by IC. The results showed that nitrite was increased during

the first 30 min of the reaction of $NO_2$ with $CaCO_3$ particles, whereas it was too little to be detected after the reaction lasted 60 min. For the reaction of $NO_2$ with the mixtures, nitrite could be detected during the all process.

At 40% RH (Fig. 1b), the absorption bands of nitrate shifted from 1040 cm$^{-1}$ to 1043 cm$^{-1}$, 746 cm$^{-1}$ to 749 cm$^{-1}$, and 816 cm$^{-1}$ to 828 cm$^{-1}$, respectively, compared to those under dry

condition. Meanwhile, the shoulder peak at 1300 cm$^{-1}$ belong to asymmetric stretching of nitrate became ambiguous. The frequency shifts of nitrate adsorption bands were caused by the phase transition of calcium nitrate. It was reported that calcium nitrate was in amorphous hydrates state at RH below 7% (Liu et al., 2008), and it deliquesced to form a saturated solution droplet at 18% RH (Tang and Fung, 1997). For the absorption bands of nitrate on the

mixtures of $CaCO_3$ and $(NH_4)_2SO_4$, there was a new shoulder peak at 1365 cm$^{-1}$ which were attributed to the $\nu_3(NO_3)$ in $NH_4NO_3$ (Schlenker et al., 2004). Moreover, the formation of $CaSO_4$ 0.5$H_2O$ was enhanced at 40% RH compared to that under dry condition, as features became apparent at 1155, 1096, and 1008 cm$^{-1}$, concomitant with the appearance of the peaks at 1620, 3555, and 3605 cm$^{-1}$ due to the vibration modes of water group in bassanite (Prasad

et al., 2005). Additionally, signatures at 1670 cm$^{-1}$, 1570 cm$^{-1}$ on the samples suggested the formation of nitric acid, $HCO_3^-$ during the heterogeneous reaction, respectively. And the signature at 1189 cm$^{-1}$ (Schlenker et al., 2004) on the mixtures suggested that $HSO_4^-$ was produced.

When RH reached 60% (Fig. 1c), water film was formed on particle surfaces with a band

centered at 1650 cm$^{-1}$ and a broad band composed of three peaks at 3260, 3400, and 3570 cm$^{-1}$, which could be assigned to the vibration modes of surface condensed water (Goodman et al., 2000). Meanwhile, the asymmetric stretching of surface nitrate appeared as a sharp peak at 1338 cm$^{-1}$. This was likely due to calcium nitrate incorporated into surface adsorbed water film and formed free aquated ions, based on the truth that only one sharp asymmetric





stretching peak existed for free aquated ions $NO_3^-$ (Gatehouse et al., 1957). The absorptions bands due to $NH_4NO_3$ could also be observed at 1365 $cm^{-1}$ for the mixtures of $CaCO_3$ and $(NH_4)_2SO_4$. Additionally, new peaks could be observed at 1168, 1145, and 1117 $cm^{-1}$, which were attributed to the $\nu_3(SO_4)$ mode of gypsum. The IR absorption bands of bassanite and

gypsum were difficult to be distinguished in the region between 1000 and 1250 $cm^{-1}$ since the $\nu_3(SO_4)$ mode of them had some overlaps in this region. gypsum showed two IR-active modes in the bending modes of crystal hydrate water at 1620 and 1685 $cm^{-1}$, while bassanite had only one band at 1620 $cm^{-1}$. And the two stretching modes of crystal hydrate water appeared at 3545, and 3400 $cm^{-1}$ for gypsum, at 3555 and 3610 $cm^{-1}$ for bassanite (Prasad,

2005). Furthermore, it should be noticed that the peak at 3400 $cm^{-1}$ from $CaSO_4 \cdot 2H_2O$ on the samples of FAS-40, FAS-57, FAS-75, and FAS-87 were much stronger than the peak at 3400 $cm^{-1}$ from condensed water on $CaCO_3$ particles. Therefore it can be inferred that $Ca(NO_3)_2$, $NH_4NO_3$, $CaSO_4 \cdot nH_2O$ (gypsum and bassanite) were produced at 60% RH from the heterogeneous reaction of $NO_2$ with the $CaCO_3$-$(NH_4)_2SO_4$ mixtures.

The spectrum of FAS-0 in Fig. 1d was similar to that in Fig. 1c, while there were considerable changes for spectra of the mixtures as RH increased to 85%. Peaks observed at 981, 998, 1131, 1177 $cm^{-1}$ on the mixtures due to the stretching vibration modes of $SO_4^{2-}$ as well as peaks at 2860, 3064, 3192 $cm^{-1}$ assigned to the stretching vibration modes of $NH_4^+$ indicated the formation of $(NH_4)_2Ca(SO_4)_2 \cdot H_2O$ (koktaite) (Jentzsch et al., 2012). The

absorption band of nitrate overlapped with that of koktaite at 749 $cm^{-1}$. It can be inferred that koktaite, an intermediate production of gypsum, was formed rapidly as a result of the interaction of ions in the liquid film after the deliquescence of surface salts (Cziczo et al., 1997; Lightstone et al., 2000). Additionally, the increasing intensity of absorption bands at 1570 $cm^{-1}$ implied that the decomposition of $CaCO_3$ was enhanced at 85% RH.

In addition, it can be concluded from Fig. 1 that $NO_2$ did not show any significant uptake on pure $(NH_4)_2SO_4$ particles (FAS-100) at all the RHs investigated. The products formed from the heterogeneous reactions of $NO_2$ with the $CaCO_3$-$(NH_4)_2SO_4$ mixtures were strongly dependent on RH. $Ca(NO_3)_2$ and bassanite were produced under both dry and wet conditions, gypsum and koktaite were formed at 60% and 85% RH.





In another set of experiments, the mixture of FAS-57 was exposed to nitrogen with corresponding RHs to investigate the solid-state reaction of $CaCO_3$ with $(NH_4)_2SO_4$ without the introduction of $NO_2$. As shown in Fig. 2, no new absorption bands occurred after exposing to dry nitrogen for 120 min. The weak peak at 1189 cm$^{-1}$ due to $HSO_4^-$ appeared as a main adsorption peak and no obvious absorption band due to bassanite could be observed on the mixture of FAS-57 at 40% RH. The results suggested that little reaction occured between $CaCO_3$ and $(NH_4)_2SO_4$ particles under dry condition and 40% RH, therefore the heterogeneous reactions of $NO_2$ with the $CaCO_3$-$(NH_4)_2SO_4$ mixtures were responsible for the formation of bassanite. Furthermore, absorption bands attributed to bassanite, gypsum, koktaite, and $HSO_4^-$ could be observed on the sample of FAS-57 after exposed to nitrogen at 60% and 85% RH. It was in good agreement with the results reported by Mori et al. (1998) that gypsum was formed from the chemical reaction between $(NH_4)_2SO_4$ and $CaCO_3$ with koktaite acting as an intermediate product. The integrated absorbance of band between 1100 and 1250 cm$^{-1}$ for the sample of FAS-57 at 60% and 85% RH in Fig. 2 was about fifty percent and seventy percent of that for FAS-57 at corresponding RH in Fig. 1. It indicated that the mixtures of $CaCO_3$ and $(NH_4)_2SO_4$ undergo obvious reaction at 60% and 85% RH without the introduction of $NO_2$. And there were additional gypsum and koktaite products formed from the heterogeneous reaction of $NO_2$ with the mixtures in comparison with the reaction between $CaCO_3$ and $(NH_4)_2SO_4$ particles under 60% and 85% RH.

## 3.2 Uptake coefficients and kinetics

The formation rates of nitrate on $CaCO_3$ particles and the mixtures were studied. The nitrate formed during the reaction was presented by the integrated absorbance ($I_A$) of the IR peak area between 1390 and 1250 cm$^{-1}$. The peak at 1043 cm$^{-1}$ was not used to avoid the interruption of the absorptions of sulfates. Figure 3 represents the integrated absorbance of nitrate as a function of time at different RHs. The nitrate formation rates were fast at initial stage, and then slowed down after a transition stage under dry condition. Moreover, the lasting time of initial stage was shortened with increasing mass fraction of $(NH_4)_2SO_4$, e.g., it lasted about 80 min for pure $CaCO_3$ particles, 20 min for the mixture of FAS-75 and 5 min for the mixture of FAS-93. The possible reasons were that active sites decreased with





increasing $(NH_4)_2SO_4$ content and the reactions occurred only on the surfaces under dry condition. While the lasting time of initial stage was extended with increasing RH, e.g., it extended to 80 min for the mixture of FAS-75, to 50 min for the mixture of FAS-93, and even longer than 120 min for the mixtures with mass fraction of $(NH_4)_2SO_4$ smaller than 57% at 40%

RH. The boundaries between initial stage and transition stage became ambiguous at 60% RH and finally disappeared at 85% RH for all the $CaCO_3$-$(NH_4)_2SO_4$ mixtures, implying that the reactions were not limited to the surface under wet conditions.

The integrated absorbance ($I_A$) for nitrate ions on the samples had a linear relationship with the amount of nitrate determined by ion chromatography $\{NO_3^-\}$:

The nitrate ions: $\{NO_3^-\}$ = (integrated absorbance $I_A$) $\times f$       (1)

Here $f$ is conversion factor. It is calculated to be $(2.11\pm0.17)\times10^{17}$ ions/int.abs at 85% RH and $(3.35\pm0.13)\times10^{17}$ ions/int.abs at 60%, 40% RH and dry condition (see Fig. S2). The conversion factor $f$ may change with the chemical environment of surface nitrate which is related to surface condensed water and ion interaction (Li et al., 2010). Then nitrate formation

rates $d\{NO_3^-\}/dt$ can be calculated from $f$ and the slope of integrated absorbance as a function of time.

As shown in Fig. 4, the initial nitrate formation rate for the samples showed a maximum value under dry condition, whereas the stable formation rates were much slower in this condition. The initial nitrate formation rates increased as RH increased from 40% RH to 60%

and 85% RH for the uptake of $NO_2$ on $CaCO_3$ particle surfaces. For the mixtures with mass fraction of $(NH_4)_2SO_4$ larger than 57%, it showed an opposite variation that initial nitrate formation rates at 40% RH were higher than that at 60% RH, followed by that at 85% RH. Besides, nitrate formation rates decreased more evidently with increasing $(NH_4)_2SO_4$ content at 85% RH and dry condition than at 40% and 60% RH, e.g., the initial nitrate formation rates

for the mixture of FAS-93 under dry condition, 40%, 60%, and 85% RH were 47%, 70%, 62%, and 34% of that for $CaCO_3$ particles at corresponding RH, respectively. It could also be demonstrated by the result that initial nitrate formation rates for $CaCO_3$ particles at 40%, 60%, and 85% RH were 64%, 67%, and 72% of that under dry condition, respectively. For the





mixture of FAS-93, the initial nitrate formation rates at 40%, 60%, and 85% RH were 95%, 87%, and 60% of that under dry condition. Similar results could be concluded from the initial nitrate formation rates of other $CaCO_3$-$(NH_4)_2SO_4$ mixtures. In summary, the initial nitrate formation rates of the reaction of $NO_2$ with $CaCO_3$-$(NH_4)_2SO_4$ mixtures were accelerated to

some degree at 40% and 60% RH in comparison with the reaction of $NO_2$ with pure $CaCO_3$ particles, whereas it was inhibited slightly at 85% RH.

The reactive uptake coefficient ($\gamma$) is defined as the rate of the reactive collisions with the surface divided by the total number of surface collisions per unit time (Z).

$$\gamma = \frac{dN(NO_2)/dt}{Z} \qquad (2)$$

$$Z = \frac{1}{4} As [NO_2] \sqrt{\frac{8RT}{\pi M_{NO_2}}} \qquad (3)$$

Where $N(NO_2)$ is the number of reactive $NO_2$ collisions with the surface, As is the effective surface area of samples and $[NO_2]$ is the gas-phase concentration of $NO_2$. R represents the gas constant, T represents the temperature and $M_{NO2}$ is the molecular weight of $NO_2$. The rate of reactive collision can be obtained from the nitrate formation rate $d\{NO_3^-\}/dt$,

then the reactive uptake coefficient can be calculated by:

$$\gamma = \frac{d\{NO_3^-\}/dt}{Z} \qquad (4)$$

The uptake coefficients of $NO_2$ on $CaCO_3$ particles and $CaCO_3$-$(NH_4)_2SO_4$ mixtures were calculated using both BET and geometric surface area, which could be considered as two extreme cases (Ullerstam et al., 2002). The results are listed in Table 2. The initial uptake

coefficients corresponding to BET surface area for $NO_2$ on $CaCO_3$ particle surfaces are $(3.34\pm0.14)\times10^{-9}$, $(2.04\pm0.07)\times10^{-9}$, $(2.23\pm0.22)\times10^{-9}$, and $(2.28\pm0.17)\times10^{-9}$ for dry condition, 40%, 60%, and 85% RH, respectively, well consistent with the previous measurement results (Li et al., 2010; Börensen et al., 2000). The $\gamma_{BET}$ is approximately a factor of $10^4$ smaller than the $\gamma_{geometric}$. The $\gamma_{BET}$ for the uptake of $NO_2$ on the mixtures was enhanced with increasing

$(NH_4)_2SO_4$ content because of the decrease of BET surface area. On the contrary, the $\gamma_{geometric}$



decreased with increasing $(NH_4)_2SO_4$ content due to the decrease of nitrate formation rate.

The mass concentrations of $NO_3^-$ formed on the samples after reaction with $NO_2$ were detected by IC, as shown in Fig. 5. The $NO_3^-$ mass concentrations for $CaCO_3$ particles are $3.22\pm0.17$, $3.31\pm0.03$, $3.38\pm0.35$, and $3.47\pm0.32$ mg/g under dry condition, 40%, 60% and 85%

RH, respectively. It suggests that the $NO_3^-$ mass concentration increase slightly with higher RH for pure $CaCO_3$ particles. For the $CaCO_3$-$(NH_4)_2SO_4$ mixtures, the $NO_3^-$ mass concentrations under dry condition are obviously smaller than those at 85% RH, and it exhibits maximum values at 40% or 60% RH. The results indicate that $CaCO_3$-$(NH_4)_2SO_4$ mixtures have a different tendency with RH compared to pure $CaCO_3$ particles. In addition, it

should be noticed that the $NO_3^-$ mass concentrations has a negative linear relation with $(NH_4)_2SO_4$ mass fraction in the mixtures under dry condition, the $R^2$ of liner fit is 0.993. Based on the product analysis, the reaction of $NO_2$ with $CaCO_3$-$(NH_4)_2SO_4$ mixtures is very similar to the reaction of $NO_2$ with pure $CaCO_3$ particles under dry condition, which indicates that the $(NH_4)_2SO_4$ has little effects on the formation of $NO_3^-$ in this condition. Moreover, the

concentrations of $NO_3^-$ of the mixtures under wet conditions are markedly larger than those under dry condition. The nitrate concentrations for the mixtures of FAS-10 and FAS-20 at 40% and 60% RH are even larger than that for pure $CaCO_3$ particles. The $NO_3^-$ mass concentrations for the mixture of FAS-57 are $3.23\pm0.09$, $3.09\pm0.14$, $2.42\pm0.07$ mg/g at 40%, 60% and 85% RH, respectively, which are increased by a factor of 2.1, 2.0, and 1.6 in

comparison with the $NO_3^-$ mass concentrations for FAS-57 under dry condition ($1.55\pm0.08$ mg/g). Similar results can be concluded from the $NO_3^-$ mass concentrations for other mixtures. Moreover, no obvious $NO_3^-$ is formed on pure $(NH_4)_2SO_4$ particles under all conditions investigated. These results clearly reveal that the mixtures exhibit a promotive effect on the heterogeneous uptake of $NO_2$ to form nitrate compared to $CaCO_3$ particles under wet

conditions.

The results described above indicate that relative humidity plays a vital role in the heterogeneous reaction of $NO_2$ with $CaCO_3$-$(NH_4)_2SO_4$ mixtures. Under dry condition, little reaction occurs between $CaCO_3$ and $(NH_4)_2SO_4$ with the absence of water vapor. Therefore, nitrate formed on the mixtures under dry condition is mainly produced from the



heterogeneous uptake of $NO_2$ on $CaCO_3$ particles without the participation of $(NH_4)_2SO_4$. At 40% RH, the solid-state reaction between $CaCO_3$ and $(NH_4)_2SO_4$ particles can be neglected, implying that solid-state reaction has little effects on the heterogeneous reaction. Meanwhile, the deliquesced $Ca(NO_3)_2$ could interact with $(NH_4)_2SO_4$ particles to form microcrystallites of

$NH_4NO_3$ and $CaSO_4·nH_2O$, which may improve the ionic mobility of the surface ions (Allen et al., 1996), modify the surface structure and expose additional active sites on $CaCO_3$ particles in the mixtures (Al-Hosney and Grassian, 2005). However, the nitrate formation rates and nitrate concentrations at 60% RH was decreased compared to those at 40% RH for the mixtures with mass percentage of $(NH_4)_2SO_4$ larger than 57%. At 60% RH, water film on

particle surfaces promotes the reaction between $CaCO_3$ and $(NH_4)_2SO_4$, leading to the formation of $CaSO_4·nH_2O$. In this condition, $CaCO_3$ particles are partly consumed during the solid-state reaction process. And $CaSO_4·nH_2O$ adhering on $CaCO_3$ particle surfaces may block active sites and inhibit the heterogeneous reaction. Consequently, the solid state reaction between $CaCO_3$ and $(NH_4)_2SO_4$ particles exhibits an inhibiting effect on the uptake

of $NO_2$ and the formation of nitrate. As for 85% RH, the deliquescence of $(NH_4)_2SO_4$ and surface nitrate leads to more water uptake on the mixture surfaces. The coagulation of ions in water film facilitates the formation of koktaite and $CaSO_4·nH_2O$. Thus, nitrate formation rate and nitrate concentration decrease at 85% RH.

**3.3 Mechanism**

According to the experimental observations described above, a reaction mechanism for the heterogeneous reactions of $NO_2$ with $CaCO_3$-$(NH_4)_2SO_4$ mixtures was proposed.

Gas phase $NO_2$ attached to surface OH groups on $CaCO_3$ particle surfaces, as shown in (R1), where (g) is the gas phase and (ads) is the adsorbed phase.

S-OH + $NO_2$(g) → S-OH…$NO_2$(ads)                                    (R1)

Börensen et al. (2000) proposed that two adsorbed-phase $NO_2$ molecules result in surface nitrate and nitrite products through a disproportionation reaction. Underwood et al. (1999b)





suggested that $NO_2$ (g) reacted to form adsorbed nitrite species initially and then react with another surface nitrite or with gas-phase $NO_2$ to form nitrate. Nitrite was detected by FTIR and IC in this study. The reaction process can be described as:

$$2\text{S-OH}\ldots NO_2(\text{ads}) \rightarrow \text{S}\ldots NO_3^-(\text{ads}) + \text{S}\ldots NO_2^-(\text{ads}) + H_2O \qquad (R2)$$

$$2\,\text{S}\ldots NO_2^-(\text{ads}) \rightarrow \text{S}\ldots NO_3^-(\text{ads}) + NO(\text{g}) \qquad (R3)$$

$$\text{S}\ldots NO_2^-(\text{ads}) + NO_2(\text{g}) \rightarrow \text{S}\ldots NO_3^-(\text{ads}) + NO(\text{g}) \qquad (R4)$$

Under dry condition, the surface nitrate was in equilibrium with surface adsorbed water and adsorbed $HNO_3$ species (R5). Adsorbed $H_2CO_3$ can exist on $CaCO_3$ particle surfaces (R6) and there was weak chemical interaction between $Ca(NO_3)_2$ and $(NH_4)_2SO_4$ (R7).

$$\text{S}\ldots NO_3^-(\text{ads}) + \text{S}\ldots H_2O(\text{ads}) \rightarrow \text{S}\ldots HNO_3(\text{ads}) + \text{S-OH} \qquad (R5)$$

$$2\text{S}\ldots HNO_3(\text{ads}) + CaCO_3 \rightarrow Ca(NO_3)_2 + \text{S}\ldots H_2CO_3(\text{ads}) \qquad (R6)$$

$$Ca(NO_3)_2 + (NH_4)_2SO_4 + 0.5H_2O \rightarrow CaSO_4\ 0.5H_2O + 2NH_4NO_3 \qquad (R7)$$

At 40% RH, $Ca(NO_3)_2$ deliquesced to form a saturated solution droplet and reacted with $(NH_4)_2SO_4$:

$$Ca^{2+} + 2NO_3^- + (NH_4)_2SO_4 + 0.5H_2O \rightarrow CaSO_4\ 0.5H_2O + 2NH_4NO_3 \qquad (R8)$$

At 60% RH, the interaction between $CaCO_3$ and $(NH_4)_2SO_4$ in the presence of surface adsorbed water film can be expressed as R9:

$$2CaCO_3 + 3(NH_4)_2SO_4 \rightarrow (NH_4)_2Ca(SO_4)_2\ H_2O + CaSO_4\ nH_2O + 4NH_3 + 2CO_2 \qquad (R9)$$

It should be noticed that $NH_3$ was detected by PTR-MS (Proton-transfer-reaction mass spectrometry) under wet conditions in this study. $NH_3$ can also be released from the decomposition of $NH_4NO_3$ (R10).

$$NH_4NO_3 \rightarrow NH_3 + HNO_3 \qquad (R10)$$

At the same time, the heterogeneous reaction of $NO_2$ with surface adsorbed water has been
demonstrated to form adsorbed $HNO_3(\text{ads})$ and gaseous HONO(g) (Svensson et al., 1987;





Jenkin et al., 1988; Goodman et al., 1999).

$$H_2O(ads) + 2NO_2(ads) \rightarrow HNO_3(ads) + HONO(g) \tag{R11}$$

At 85% RH, the interaction of ions in the water film can be expressed as:

$$2Ca^{2+} + 3SO_4^{2-} + 2NH_4^+ + nH_2O \rightarrow CaSO_4 \cdot nH_2O + (NH_4)_2Ca(SO_4)_2 \cdot H_2O \tag{R12}$$

## 4. Conclusions and atmospheric implications

The surface products and kinetics of the heterogeneous reactions of $NO_2$ with pure $CaCO_3$ particles, pure $(NH_4)_2SO_4$ particles, and $CaCO_3$-$(NH_4)_2SO_4$ mixtures were investigated under various RHs, using DRIFTS technique. And the solid-state reaction between $CaCO_3$ and

$(NH_4)_2SO_4$ particles were studied for comparison. All these reactions can occur in practical atmospheric conditions, which can be expressed in Fig. 6. The findings in this study have important atmospheric implications.

Calcium nitrate was produced from the heterogeneous reaction of $NO_2$ with $CaCO_3$-$(NH_4)_2SO_4$ mixtures under both dry and wet conditions, and bassanite, gypsum and

koktaite were formed depending on RH. It suggested that chemical composition in particulate phase was changed during the heterogeneous process, which can affect the physicochemical characteristics of atmospheric particles, including hygroscopicity, optical properties, and chemical reactivity. Besides, koktaite was detected in aerosols collected in Beijing, while it was absent in the soil where the Asian dust originates (Mori et al., 2003), large uncertainties

remain about its formation in the atmosphere. The results presented here provide evidence that the heterogeneous reactions of mixed $CaCO_3$-$(NH_4)_2SO_4$ particles with atmospheric acid trace gases was a possible source of koktaite. Also, the results indicated that the uptake of $NO_2$ and the formation of nitrate promoted removing $SO_4^{2-}$ from water soluble species such as $(NH_4)_2SO_4$ to insoluble gypsum species, which could reduce the atmospheric water soluble

sulfate content.

Gas phase products such as $NH_3$ could be released during the heterogeneous reaction of $NO_2$ with $CaCO_3$-$(NH_4)_2SO_4$ mixtures. In the atmosphere $NH_3$ is mainly emitted from





agriculture activities (such as fertilization and animal feeding) and biomass burning, and it plays an important role in nucleation and the growth of ion cluster and nanoparticles. The results in this study suggest that heterogeneous uptake of $NO_2$ on $CaCO_3$ particles with the presence of $(NH_4)_2SO_4$ may be a potential pathway for the transformation of $NH_3$ from

particulate phase to gas phase.

Furthermore, the uptake-coefficients of $NO_2$ on $CaCO_3$-$(NH_4)_2SO_4$ mixtures were determined, providing kinetic data for modeling studies. The results illustrate that the presence of $(NH_4)_2SO_4$ exhibits a promotive effect on the nitrate formation under wet conditions as a result of the interaction between $Ca(NO_3)_2$ and $(NH_4)_2SO_4$. On the contrary,

the reaction between $CaCO_3$ and $(NH_4)_2SO_4$ particles has an inhibiting effect on the formation of nitrate during the heterogeneous reaction process, especially at high RH. Considering the abundance of $(NH_4)_2SO_4$ in the atmospheric aerosols, its mixtures with mineral dust may significantly affect nitrate formation and the content of nitrate in atmospheric particles. The multicomponent reaction systems under ambient RH conditions

play as yet unclear but potentially vital role in atmospheric processes. To better understand the role of heterogeneous reactions in the atmospheric chemistry, the effects of ambient RH as well as multicomponent reaction systems should be considered.

**The Supplement related to this article is available online.**

*Author contributions.* Fang Tan and Shengrui Tong contribute equally to this work.

*Acknowledgements.* This project was supported by the Strategic Priority Research Program (B) of the Chinese Academy of Sciences (Grant No. XDB05010400), and the National Natural Science Foundation of China (Contract No.41475114, 91544227, 21477134).

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



**Table 1.** Assignments of IR vibration frequencies of surface adsorbed species formed on $CaCO_3$ particle surfaces and $CaCO_3$-$(NH_4)_2SO_4$ mixtures

| Samples | Stretch mode | $\nu_1$ $(cm^{-1})$ | $\nu_2$ $(cm^{-1})$ | $\nu_3$ $(cm^{-1})$ | $\nu_4$ $(cm^{-1})$ | Stretch mode | |
|---|---|---|---|---|---|---|---|
| $Ca(NO_3)_2$ | $NO_3^-$ | 1043 | 816 | 1300, 1330, | 748 | | |
| [a]$NH_4NO_3$ | $NO_3^-$ | 1050 | 830 | 1333, 1365 | | $NH_4^+$ | 1426, 1451 |
| $CaSO_4$ $0.5H_2O$ | $SO_4^{2-}$ | 1008 | | 1096, 1116, 1155, 1168 | | $H_2O$ | 1620, 3553, 3605 |
| [b]$CaSO_4$ $0.5H_2O$ | $SO_4^{2-}$ | 1008 | | 1096, 1116, 1153, 1168 | 601, 660 | $H_2O$ | 1620, 3550, 3610 |
| $CaSO_4$ $2H_2O$ | $SO_4^{2-}$ | 1003 | | 1117, 1145, 1167 | | $H_2O$ | 1620, 1685, 3400, 3545 |
| [c]$CaSO_4$ $2H_2O$ | $SO_4^{2-}$ | 1005 | | 1117, 1145, 1167 | 602, 669 | $H_2O$ | 1621, 1685, 3405, 3495, 3547 |
| $(NH_4)_2Ca(SO_4)_2$ $H_2O$ | $SO_4^{2-}$ | 981, 998 | | 1131, 1177 | | $H_2O$ | 2860, 3064, 3192 |
| [d]$(NH_4)_2Ca(SO_4)_2$ $H_2O$ | $SO_4^{2-}$ | 981, 998 | | 1108, 1131, 1177 | 602, 614, 646, 656 | $H_2O$ | 2857, 2922, 3125, 3192 |

[a] from Schlenker et al. (2004). [b,c] from Prasad et al. (2005). [d] from Jentzsch et al. (2012)





**Table 2.** Initial uptake coefficients calculated using BET surface area and geometric surface area for $NO_2$ on $CaCO_3$ particle surfaces and $CaCO_3$-$(NH_4)_2SO_4$ mixtures at various RHs.

| $(NH_4)_2SO_4$ (wt%) | dry condition | | 40% RH | | 60% RH | | 85% RH | |
|---|---|---|---|---|---|---|---|---|
| | $\gamma_{BET}$ ($\times10^{-9}$) | $\gamma_{geo}$ ($\times10^{-6}$) | $\gamma_{BET}$ ($\times10^{-9}$) | $\gamma_{geo}$ ($\times10^{-6}$) | $\gamma_{BET}$ ($\times10^{-9}$) | $\gamma_{geo}$ ($\times10^{-6}$) | $\gamma_{BET}$ ($\times10^{-9}$) | $\gamma_{geo}$ ($\times10^{-6}$) |
| 0 | 3.34±0.14 | 10.4±0.44 | 2.04±0.07 | 6.36±0.22 | 2.23±0.22 | 6.94±0.69 | 2.28±0.17 | 7.10±0.53 |
| 10 | 3.19±0.21 | 9.83±0.65 | 2.06±0.21 | 6.34±0.45 | 2.25±0.14 | 6.91±0.43 | 2.13±0.41 | 6.56±1.26 |
| 20 | 3.77±0.24 | 9.54±0.61 | 2.51±0.34 | 6.28±0.86 | 2.74±0.42 | 6.87±1.06 | 2.00±0.21 | 5.63±0.53 |
| 40 | 5.34±0.17 | 9.25±0.29 | 3.50±0.42 | 6.07±0.72 | 3.67±0.48 | 6.36±0.83 | 3.15±0.28 | 5.46±0.49 |
| 57 | 6.82±0.33 | 8.38±0.41 | 4.70±0.51 | 5.78±0.63 | 4.47±0.26 | 5.49±0.32 | 4.15±0.53 | 5.10±0.65 |
| 75 | 7.74±0.94 | 6.94±0.84 | 6.12±0.37 | 5.49±0.23 | 5.80±0.53 | 5.20±0.48 | 4.26±0.31 | 3.82±0.28 |
| 87 | 9.04±0.73 | 5.78±0.46 | 7.68±0.50 | 4.92±0.32 | 7.22±0.63 | 4.63±0.40 | 4.83±0.46 | 3.10±0.19 |
| 93 | 14.4±1.07 | 4.90±0.36 | 13.6±0.93 | 4.63±0.32 | 12.7±0.81 | 4.34±0.28 | 7.48±0.82 | 2.55±0.28 |





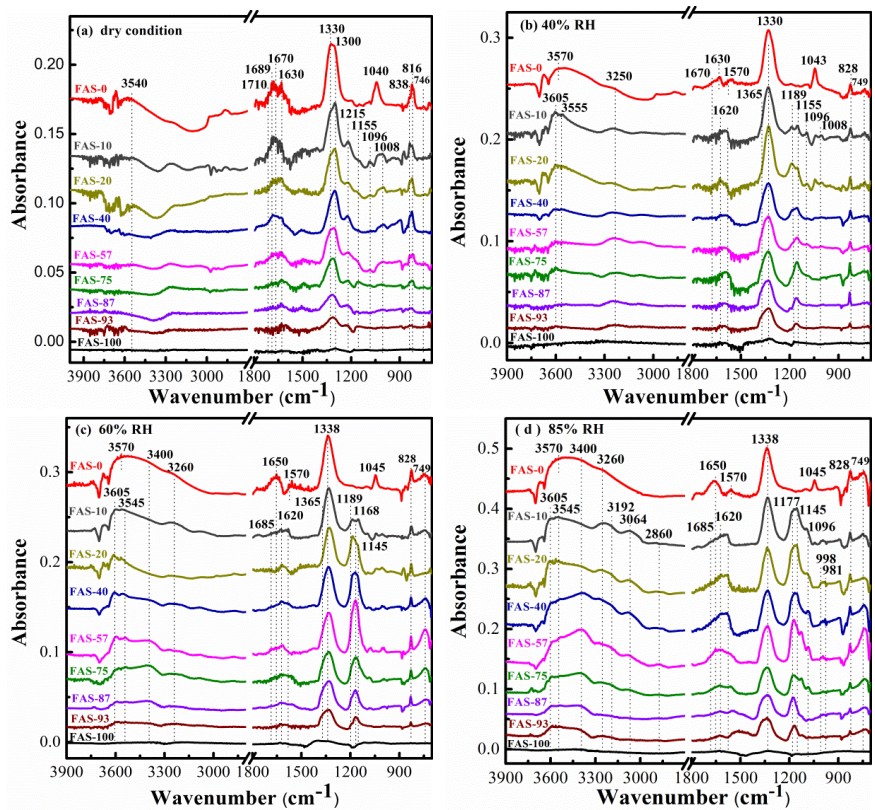

**Figure 1.** DRIFTS spectra of CaCO$_3$ particles (FAS-0), CaCO$_3$-(NH$_4$)$_2$SO$_4$ mixtures (FAS-10 - FAS-93), and (NH$_4$)$_2$SO$_4$ particles (FAS-100) after reaction with NO$_2$ at (a) dry condition, (b) 40% RH, (c) 60% RH, (d) 85% RH for 120 min. NO$_2$ concentration was $2.6 \times 10^{15}$ molecule cm$^{-3}$.





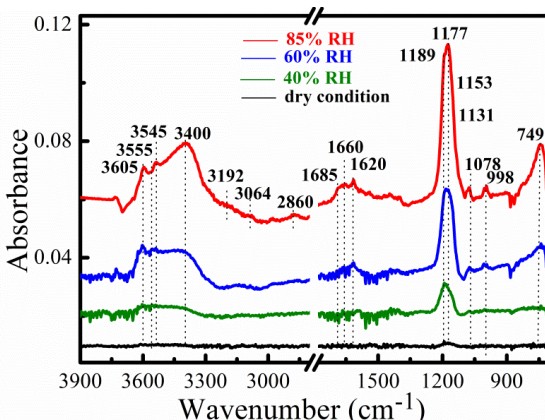

**Figure 2.** In situ DRIFTS spectra of surface products when the mixture of FAS-57 were exposed to nitrogen at dry condition (black), 40% RH(green), 60% RH(blue) and 85% RH (red) for 120 min.





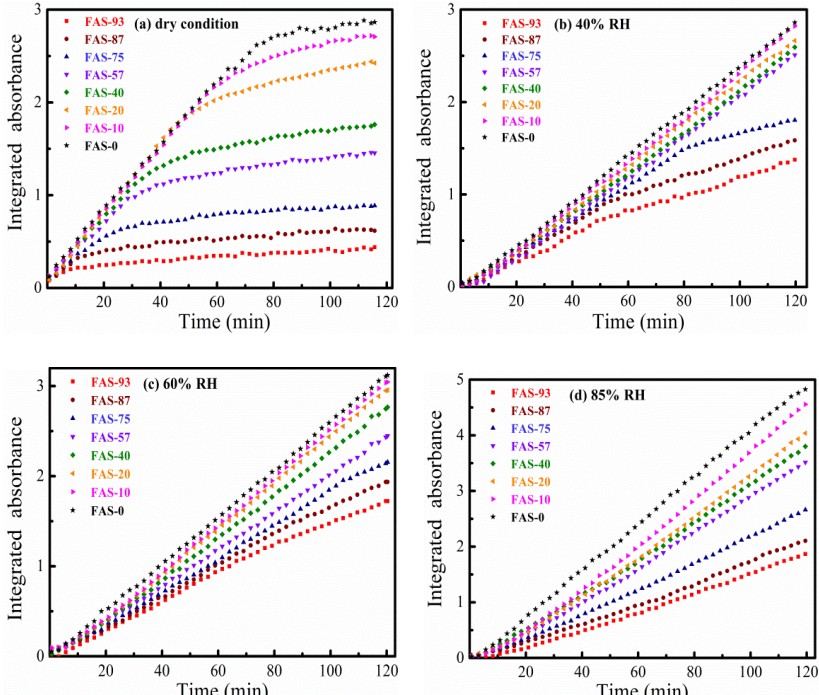

**Figure 3.** The integrated absorbance of the peak area between 1390 and 1250 $cm^{-1}$ for nitrate on pure $CaCO_3$ particle surfaces (FAS-0), and $CaCO_3$-$(NH_4)_2SO_4$ mixtures (FAS-10 - FAS-93) at (a) dry condition, (b) 40% RH, (c) 60% RH, and (d) 85% RH. The $NO_2$ concentration was $2.6\times10^{15}$ molecule $cm^{-3}$.





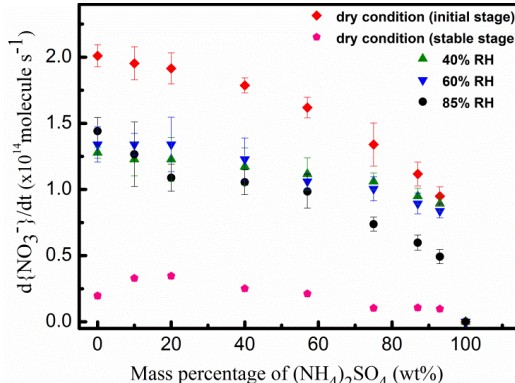

**Figure 4**. Initial nitrate formation rates at dry condition (rhombus), 40% RH (triangle), 60% RH (fall triangle), 85% RH (roundness) and stable nitrate formation rate (pentagon) under dry condition versus the mass percentage of $(NH_4)_2SO_4$ in the mixtures. The data points and the error bars are the average value and the standard deviation of three duplicate experiments.



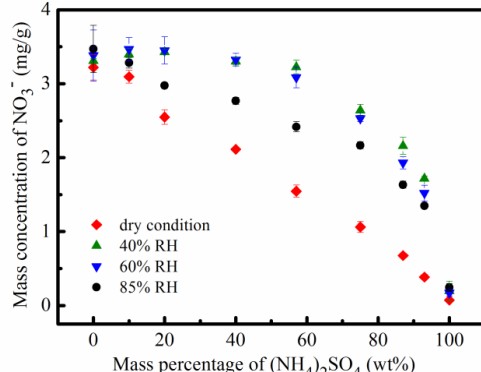

**Figure 5**. The mass concentration of $NO_3^-$ for $CaCO_3$ particles and the $CaCO_3$-$(NH_4)_2SO_4$ mixtures after reacted with $NO_2$ for 120 min as a function of the mass percentage of $(NH_4)_2SO_4$ in the mixtures. The data points and the error bars are the average value and the standard deviation of three duplicate experiments.





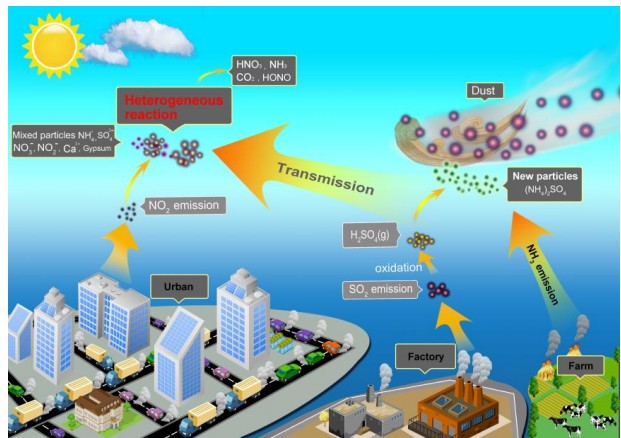

**Figure 6.** Schematic illustrating the possible heterogeneous processes of $NO_2$ with $CaCO_3$-$(NH_4)_2SO_4$ mixtures and the possible atmospheric implications.