# Peer review of "Heterogeneous reactions of NO2 with CaCO3-(NH4)2SO4 mixtures at different relative humidities"

_Atmospheric Chemistry and Physics, 2016_

## Referee Comment (RC1) · Anonymous Referee #2 · 14 Apr 2016

General comments: The article aims to understand the uptake and kinetic behavior of a mixed aerosols system with its reaction with NO2. The article has laid out all the aspects of the experiments and presented the data well. The role of (NH4)2SO4 in the reaction was analyzed well. The data, summaries and mechanisms fits well but there are few major contradictory statements made in the different sections of the article that need clarification:

I recommend publication after a rewrite clarifying some of the major contradictory statement highlighted below:

Specific comments

The main issue I have is the role of (NH4)2SO4 in the reaction. There seems to be two contradictory summaries being presented here, without explanation on how/why

the (NH4)2SO4 is causing these effects. There seems to be a cutoff RH value (60%), below which the effect of (NH4)2SO4 is promotive and above which the effect is opposite (see page 10, line 26; page 12, line 3; page 13, line 19, or 21 ;). The authors have proposed active site dependence, (page 10, line 26) and deliquescence of (NH4)2SO4 (page 14, line 16) as possible reasons for this. The way the sample mixture was made (page 5 line 21) contradicts the first reason; and these negative effect starts at 60% RH (which is further lower that DRH of (NH4)2SO4 contradicts the second reason. The role of (NH4)2SO4 is important (as the authors have clearly shown), their reasons for the observed effects need more explanations, and these contradictory statements do not help the reader/article.

Page 9, line 2-5: The identification of CaSO4.0.5H2O and CaSO4.2H2O uses very similar IR peaks. It's not entirely clear how these same peaks were used to differentiate the CaSO4.0.5H2O from the CaSO4.2H2O.

Page 9 line 21: How is the decomposition of CaCO3 manifest itself as an increasing intensity of the 1570 cm-1 band? Decomposition usually leads to a negative (loss of) intensity, not a positive (increasing) intensity. The 1570 cm-1 has been assigned to HSO4-, how is the increasing intensity of this peak tie-in to the loss of CaCO3? I am assuming it's from a specific reaction, but this is not clearly stated here.

Page 9 line 27: "…surface nitrate was decreased with increased Ca(NO3)2 content..". The sentence seems contradictory, how was the surface nitrate and bulk nitrate differentiated from the spectra?

Page 10 line 14-15: "…was faster than the reaction of…" how was this (fast reaction) determined? Needs more explanation.

Page 12, equation 2 and 3: Why are there two formulae for the calculation of reactive uptake coefficient? One uses dN(NO2) and the other uses dNO3?

Page 14, line 1 : They report that the amorphous hydrate Ca(NO3)2 has weak interaction with (NH4)2SO4, but the following sentence ( same page, line 5), they suggest that Ca(NO3)2 could interact with (NH4)2SO4 to form NH4NO3. How do they explain these contradictory statements?

Technical corrections:

Page 3 line 2: should be "gaseous", not gases.

Page 3 line 8 : remove the "1" in front of "Pathak".

Page 3 line 16: "...significant relevance". Incomplete sentences, relevance to what?

Page 3, line 22: change "was" to "were".

Page 3 line 26: "...after being exposed to...".

Page 4 line 4: "...attributing it to...".

Page 4 line 12: what do they mean by, "The catalysis and basic coexists could...."

Page 6 line 9: add "respectively" at the end of the sentence.

Page 7 line 7: How "dry" (< 1% humidity?) were the experimental conditions? It has been reported in literature that there are enough water layers at RH <5% RH to influence surface reactions. Their "dry" experimental (RH) conditions should be presented.

Page 9 line 23: "...can be concluded..."

Page 10 line 1: "...add a comma after N2, its confusing without it.

Page 11, line 17: where are the "...stable formation" states/rates? This statement needs to be explained.

Page 12, line 10: create a better notation for the effective surface area, because "As" is confusing.

Page 13, line 29: remove "with absence of water vapor". It's redundant since you have

mentioned "under dry conditions" at the beginning of the sentence.

Page 14, line 9: ". . .decreased. . ."

---

## Referee Comment (RC2) · Anonymous Referee #1 · 19 Apr 2016

Mineral dust and sulfate are common components in atmospheric particulate maters (PMs), and their coagulation in the atmosphere can form new types of PMs whose physical and chemical characters will be altered, thus affecting on atmospheric physical and chemical processes. Because only few studies investigated the heterogeneous reactions under complex conditions, there is still large gap to explain many phenomena of field measurements by using the current knowledge of atmospheric chemistry. The new finding in this study about the heterogeneous reactions of NO2 on the surface of CaCO 3-(NH 4 ) 2 SO 4 mixtures provided important information, that is, the heterogeneous reactions in the atmosphere may play important role on formation of nitrate, CaSO 4  0.5H 2 O (bassanite), CaSO 4  2H 2 O (gypsum) and (NH 4 ) 2 Ca(SO 4 ) 2  H 2 O (koktaite). This reviewer recommends the manuscript be published in the journal after considering the following one specific: The formation rates of nitrate

based on the integrated absorbance of the IR peak area between 1390 and 1250 cm -1 are inconsistent with the final concentrations of nitrate measured by IC. The intensity of the IR absorbance in the DRIFTS can only reveal the surface concentration of nitrate, whereas the nitrate concentrations measured by IC are the bulk concentrations in the PMs. The surface nitrate formed through the heterogeneous reactions was suspected to easily diffuse into inner layer of the PMs under high RH conditions. The authors should present the explanations.

---

## Author Comment (AC1)

**Response**

**Response to referee:**

We are grateful to Referee #2 for giving valuable comments and helpful suggestions to improve our manuscript. Our response to the comments and changes to the manuscript are included below. We repeat the specific points raised by the reviewer in bold font, followed by our response in italic font. The manuscript that referee #2 commented is the version that delivered to Atmospheric Chemistry and Physics (ACP) initially, while we had made some small modifications according to the suggestions from Referee #1 in the ACPD version. The numbers of pages and lines are consistent with those in the ACPD paper to avoid misunderstanding.

General comments: The article aims to understand the uptake and kinetic behavior of a mixed aerosols system with its reaction with NO2. The article has laid out all the aspects of the experiments and presented the data well. The role of  $(NH_4)_2SO_4$  in the reaction was analyzed well. The data, summaries and mechanisms fits well but there are few major contradictory statements made in the different sections of the article that need clarification.

I recommend publication after a rewrite clarifying some of the major contradictory statement highlighted below:

**Reply:** We appreciate the reviewer's comments. And we have carefully revised our manuscript according to the reviewer's suggestions.

**Specific comments**

1. The main issue I have is the role of  $(NH_4)_2SO_4$  in the reaction. There seems to be two contradictory summaries being presented here, without explanation on how/why the  $(NH_4)_2SO_4$  is causing these effects. There seems to be a cutoff RH value (60%), below which the effect of  $(NH_4)_2SO_4$  is promotive and above which the effect is opposite (see page 10, line 26; page 12, line 3; page 13, line 19, or 21 ;). The authors have proposed active site dependence, (page 10, line 26) and deliquescence of  $(NH_4)_2SO_4$  (page 14, line 16) as possible reasons for this. The way the sample mixture was made (page 5 line 21) contradicts the first reason; and these negative effect starts at 60% RH (which is further lower that DRH of  $(NH_4)_2SO_4$  contradicts the second reason. The role of  $(NH_4)_2SO_4$  is important (as the authors have clearly shown), their reasons for the observed effects need more explanations, and these contradictory statements do not help the reader/article.

**Reply:** Thanks for the reviewer's comment. We were regretful that we did not clarify enough about how  $(NH_4)_2SO_4$  was causing effects in the heterogeneous reaction, resulting in misunderstanding of the reviewer and reader. In fact, we did not mention that active site dependence and the deliquescence of  $(NH_4)_2SO_4$  were responsible for the effects of  $(NH_4)_2SO_4$  in this paper. And 60% RH is not a cutoff RH between the two opposite effects. We did emphasize that the chemical interaction of  $(NH_4)_2SO_4$  with  $Ca(NO_3)_2$  or  $CaCO_3$  were the possible reasons for the promotive or inhibiting effects (page 10 line 1-19, page 13 line 12-14, page 13 line 27-29, page 14 line 1-15) and the nitrate concentrations were enhanced under all the wet conditions investigated (40%, 60% and 85% RH) (page 13 line 14-21 and line 23-25).

Firstly,  $(NH_4)_2SO_4$  has little effects on nitrate formation in the heterogeneous reaction of the mixtures with NO2 under dry condition (page 13 line 27-29 and page 14 line 1). Figure 1a indicated that  $(NH_4)_2SO_4$  particles has limited interaction with the amorphous state  $Ca(NO_3)_2$  and Figure 2 suggested that it has little reaction with  $CaCO_3$  particles under dry condition. The reactive sites dependence was the possible reason to explain the results that the lasting time of initial stages and the  $NO_3^-$  mass concentrations decrease linearly with increasing  $(NH_4)_2SO_4$  content in the mixtures (as the reviewer mentioned, on page 10 line 27-28 in initial manuscript version), since the nitrate is produced from the uptake of  $NO_2$  on  $CaCO_3$  particles without the participation of  $(NH_4)_2SO_4$  under dry condition.

As RH increased from dry condition to 40% RH, the chemical reaction of  $CaCO_3$ with  $(NH_4)_2SO_4$  particles is still neglectable (Figure 2). And the chemical interaction of the deliquesced  $Ca(NO_3)_2$  with  $(NH_4)_2SO_4$  particles are responsible for the formation of  $NH_4NO_3$  and  $CaSO_4 0.5H_2O$ , which may enhance the ionic mobility of the surface ions (Allen et al., 1996), modify the surface structure and re-expose reactive sites (Al-Hosney and Grassian, 2005), consequently showing promotive effects on the nitrate formation during the heterogeneous reaction of  $NO_2$  with the mixtures.

At 60% RH, a chemical reaction in the coagulation of  $CaCO_3$  and  $(NH_4)_2SO_4$ particles actually occurs without the introduction of NO2 (page 10 line 9-11). Consequently,  $CaCO_3$  particles are partly consumed during the coagulation with  $(NH_4)_2SO_4$  and the CaSO4 nH2O formed in the coagulation may block reactive sites for further reaction, resulting in an inhibiting effect on nitrate formation (page 14 line 9-15). At the same time, the deliquesced  $Ca(NO_3)_2$  still has chemical interactions with  $(NH_4)_2SO_4$  (page 10 line 13-19). Therefore, there is a combined effect of the two opposing effects from the interaction of  $(NH_4)_2SO_4$  with  $Ca(NO_3)_2$  and the interaction of (NH4)2SO4 with CaCO3. Furthermore, it is well consistent with the results that the nitrate formation rates and the  $NO_3^-$  mass concentrations at 60% RH are slightly larger than those at 40% RH for the mixtures with mass percentage of  $(NH_4)_2SO_4$  smaller than 43%, while it is opposite for the mixtures with mass percentage of  $(NH_4)_2SO_4$  larger than 57%. Thus 60% RH is not a cutoff value. As for 85% RH, the deliquescence of  $(NH_4)_2SO_4$  (Cziczo et al., 1997) leads to more water uptake on the mixture surfaces, facilitating the reaction of  $(NH_4)_2SO_4$  with CaCO3 (page 14 line15-16). Therefore the negative effects are more obvious at 85% RH than at 60% and 40% RH. It should be noticed that although the nitrate formation rates and NO3- mass concentrations at 85% RH are smaller than those at 60% and 40% RH, the nitrate concentrations are still improved at 85% RH (page 13 line 17-25). Some modifications have been made in order to clarify clearly how  $(NH_4)_2SO_4$  affects the nitrate formation in the heterogeneous reaction of  $NO_2$  with  $CaCO_3$ -( $NH_4$ )2SO4

mixtures at different RHs.

**Related changes included in the revised manuscript:**

Page 2 line 15-17: the sentence "Under wet conditions, the  $CaCO_3$ - $(NH_4)_2SO_4$ mixtures exhibited...." was revised to "Under wet conditions, the chemical interaction of  $(NH_4)_2SO_4$  with  $Ca(NO_3)_2$  has a promotive effects on the nitrate formation in the heterogeneous reaction of the mixtures with NO2, while the coagulation of  $(NH_4)_2SO_4$ with CaCO3 exhibits an inhibiting effects at the same time. The nitrate formation is promoted in the heterogeneous reaction of NO2 with CaCO3- $(NH_4)_2SO_4$  mixtures, especially at medium RHs."

Page 10 line 7-9: the sentence "... therefore the heterogeneous reactions of  $NO_2$  with the  $CaCO_3$ - $(NH_4)_2SO_4$  mixtures were responsible for the formation of bassanite." was revised to "... therefore the chemical interaction of  $Ca(NO_3)_2$  with  $(NH_4)_2SO_4$  was responsible for the formation of bassanite in these conditions."

Page 10 line 17-19: the sentence "And there were additional gypsum and koktaite products formed...." was revised to "Thus  $CaSO_4 nH_2O$  and koktaite products could be formed both from the chemical interaction of  $(NH_4)_2SO_4$  with  $Ca(NO_3)_2$  and the reaction of  $(NH_4)_2SO_4$  with  $CaCO_3$  at 60% and 85% RH."

Page 11 line 22, after the sentence "...followed by that at 85% RH" we added "While for the mixtures with mass fraction of  $(NH_4)_2SO_4$  smaller than 43%, the nitrate formation rates increased initially as RH elevated from 40% RH to 60% RH then it decreased obviously as RH increased to 85% RH. The differences in the tend of the nitrate formation rates with RH for the mixtures could be explained by the combined opposite effects from the interaction of  $(NH_4)_2SO_4$  with  $Ca(NO_3)_2$  or  $CaCO_3$  at 60% RH."

Page 11 line 26, after the sentence "...at corresponding RH, respectively" we added "As RH increased from dry condition to 40% and 60% RH, the initial nitrate formation rates decreased less for the reaction of NO2 with the mixtures than with CaCO3 particles, while it was opposite as RH increased to 85% RH"

Page 13 line 17-18: before the sentence "The  $NO_3^-$  mass concentrations for the mixture of FAS-57..." we added "The  $NO_3^-$  mass concentrations increase much more for the mixtures than for pure CaCO3 particles as RH elevated from dry condition to wet conditions, e.g."

Page 14 line 7: after the sentence "...expose additional active sites on  $CaCO_3$  particles in the mixtures" we added "Thus the chemical interaction of  $Ca(NO_3)_2$  and

 $(NH_4)_2SO_4$  particles may exhibits promotive effects on the nitrate formation during the heterogeneous reaction of NO2 with CaCO3-(NH4)2SO4 mixtures."

Page 14 line 7-9: the sentence "However, the nitrate formation rates and nitrate concentrations at 60% RH was decreased compared to those at 40% RH for the mixtures with mass percentage of  $(NH_4)_2SO_4$  larger than 57%." was revised to "The nitrate formation rates and nitrate concentrations increase slightly when RH increased from 40% RH to 60% RH for the mixtures with mass percentage of  $(NH_4)_2SO_4$  less than 43%. However, it was opposite for the mixtures with mass percentage of nitrate formation rates and nitrate concentrations at 60% RH are smaller than those at 40% RH."

References:

Al-Abadleh, H. A., Al-Hosney, H. A., and Grassian, V. H.: Oxide and carbonate surfaces as environmental interfaces: the importance of water in surface composition and surface reactivity, J. Mol. Catal. A: Chem., 228, 47-54, doi:10.1016/j.molcata.2004.09.059, 2004.

Al-Hosney, H. A., and Grassian, V. H.: Water, sulfur dioxide and nitric acid adsorption on calcium carbonate: A transmission and ATR-FTIR study, Phys. Chem. Chem. Phys., 7, 1266-1276, doi:10.1039/b417872f, 2005.

Allen, H. C., Laux, J. M., Vogt, R., Finlayson-Pitts, B. J., and Hemminger, J. C.: Water-induced reorganization of ultrathin nitrate films on NaCl: Implications for the tropospheric chemistry of sea salt particles, J. Phys. Chem., 100, 6371-6375, doi:10.1021/jp953675a, 1996.

Mori, I., Nishikawa, M., and Iwasaka, Y.: Chemical reaction during the coagulation of ammonium sulphate and mineral particles in the atmosphere, Sci. Tot. Environ., 224, 87-91, doi:10.1016/s0048-9697(98)00323-4, 1998.

2. Page 9, line 2-5: The identification of CaSO4.0.5H2O and CaSO4.2H2O uses very similar IR peaks. It's not entirely clear how these same peaks were used to differentiate the CaSO4.0.5H2O from the CaSO4.2H2O.

Reply: Thanks for the reviewer's comment. The IR absorption peaks at 1008 and 1116

cm-1 due to CaSO4.0.5H2O and the peaks at 1005 and 1117 cm-1 due to CaSO4.2H2O are hard to distinguish. There are, actually, some features that can be used to differentiate CaSO4.0.5H2O from CaSO4.2H2O. As has been described in this paper (page 9, line 6-9), the peaks at 1096 and 1155 cm-1 belong to CaSO4.0.5H2O can be clearly observed in the IR spectrum, which are evidences for the formation of CaSO4.0.5H2O rather than CaSO4.2H2O. Besides, CaSO4.2H2O shows two IR-active modes in the bending modes of crystal hydrate water at 1620 and 1685 cm-1, while CaSO4.0.5H2O has only one band at 1620 cm-1. Furthermore, the two stretching modes of crystal hydrate water occur at 3495, 3545 and 3400 cm-1 for CaSO4.2H2O, at 3555 and 3610 cm-1 for CaSO4.0.5H2O (Prasad, 2005; Liu et al., 2009).

**Related changes included in the revised manuscript:**

Page 9 line 4-6: the sentence "The IR absorption bands of …" was revised to "Although the IR absorption bands of bassanite and gypsum had some overlaps in the region between 1000 and 1250 cm-1, there were some features that could be used to differentiate  $CaSO_4.0.5H_2O$  from  $CaSO_4.2H_2O$ ."

References:

Liu, Y., Wang, A., Freeman, J. J.: Raman, Mir, and NIR spectroscopic study of calcium sulfates: gypsum,bassanite, and anhydrite, 40th Lunar and Planetary Science Conference, 2009.

Prasad, P. S. R., Krishna Chaitanya, V., Shiva Prasad, K., and Narayana Rao, D.: Direct formation of the γ-CaSO4 phase in dehydration process of gypsum: In situ FTIR study, Am. Mineral., 90, 672-678, doi:10.2138/am.2005.1742, 2005.

**3.** Page 9 line 21: How is the decomposition of CaCO3 manifest itself as an increasing intensity of the 1570 cm-1 band? Decomposition usually leads to a negative (loss of) intensity, not a positive (increasing) intensity. The 1570 cm-1 has been assigned to HSO4-, how is the increasing intensity of this peak tie-in to the loss of CaCO3? I am assuming it's from a specific reaction, but this is not clearly stated here.

Reply: Thanks for the reviewer's suggestions. Normally, the decomposition of

reactants leads to a negative intensity of IR spectrum in DRIFTS experiments. In this study, the IR absorption peak at 1570 cm-1 is assigned to the asymmetric stretching of HCO3- (Al-Hosney et al., 2004; Li et al., 2010). In fact, there is no interruption from the IR absorption bands of other reactants and products in this range. The positive intensity is likely due to the increasing information of HCO3-, which is from the decomposition of bulk CaCO3 under wet conditions. As indicated in Figure 1, the peak at 1570 cm-1 did not appear under dry condition and it increased with increasing RH. The reactions are limited to surfaces and H2CO3 can exist as absorbed phase under dry condition. While the reaction of NO2 can occur not only on the surfaces of CaCO3 and mixtures but also into the bulk of the samples in the presence of surface condensed water (Goodman et al., 2001; Goodman et al., 2000). Furthermore, the acidity of surface condensed water is enhanced as a result of the formation of HNO3 and the dissolution of (NH4)2SO4, which facilitates the decomposition of the bulk CaCO3 particles.

**Related changes included in the revised manuscript:**

Page 9 line 23-24: the sentence "Additionally, the increasing intensity of absorption bands at 1570 cm-1 implied that the decomposition of CaCO3 was enhanced at 85% RH." was revised to "Additionally, the IR absorption peaks at 1570 cm-1 in Figure 1d were much stronger than those at 40% and 60% RH. The positive intensity was likely due to the increasing information of  $HCO_3^-$ , which was from the decomposition of the bulk CaCO3 under wet conditions. It could be interpreted that the reaction of NO2 can occur not only on the surfaces of CaCO3 and the mixtures but also into the bulk of the samples under wet conditions. Also the acidity of surface condensed water was enhanced as a result of the formation of HNO3 and the dissolution of  $(NH_4)_2SO_4$ , which facilitates the decomposition of bulk CaCO3 particles."

**References:**

Al-Hosney, H. A., and Grassian, V. H.: Water, sulfur dioxide and nitric acid adsorption on calcium carbonate: A transmission and ATR-FTIR study, Phys. Chem. Chem. Phys., 7, 1266-1276, doi:10.1039/b417872f, 2005. Al-Hosney, H. A., and Grassian, V. H.: Carbonic Acid: an important intermediate in the surface chemistry of calcium carbonate, J. Am. Chem. Soc., 126, 8068-8069, doi:10.1021/ja0490774, 2004.

Goodman, A. L., Bernard, E. T., and Grassian, V. H.: Spectroscopic study of nitric acid and water adsorption on oxide particles: Enhanced nitric acid uptake kinetics in the presence of adsorbed water, J. Phys. Chem. A, 105, 6443-6457, doi:10.1021/jp0037221, 2001.

Goodman, A. L., Underwood, G. M., and Grassian, V. H.: A laboratory study of the heterogeneous reaction of nitric acid on calcium carbonate particles, J. Geophys. Res., 105, 29053-29064, doi:10.1029/2000jd900396, 2000.

Li, H. J., Zhu, T., Zhao, D. F., Zhang, Z. F., and Chen, Z. M. : Kinetics and mechanisms of heterogeneous reaction of NO2 on CaCO3 surfaces under dry and wet conditions, Atmos. Chem. Phys., 10, 463–474, 2010.

4. Page 9 line 27: "...surface nitrate was decreased with increased  $Ca(NO_3)_2$  content..". The sentence seems contradictory, how was the surface nitrate and bulk nitrate differentiated from the spectra?

**Reply:** Thanks for the reviewer's advice. This sentence should be corrected to "Moreover, the surface nitrate was decreased with increasing  $(NH_4)_2SO_4$  content in mixtures." (page 9 line 27-28). This sentence was in the initial manuscript and it had been deleted in the ACPD version. I think it cannot differentiate surface nitrate from bulk nitrate according to IR spectra.

**5.** Page 10 line 14-15: "...was faster than the reaction of..." how was this (fast reaction) determined? Needs more explanation.

**Reply:** Thanks for the reviewer's advice. We realized that the sentence "This is likely due to the fact that the reaction between  $(NH_4)_2SO_4$  and  $Ca(NO_3)_2$  was faster than the reaction of  $(NH_4)_2SO_4$  with CaCO\_3." was misleading. What we wanted to express was that  $Ca(NO_3)_2$  were more hygroscopic and soluble than CaCO\_3 and it may has stronger chemical interaction with  $(NH_4)_2SO_4$  than CaCO\_3 particles under the same condition.

This sentence was in the initial manuscript and it had been deleted in the ACPD version.

**Related changes included in the revised manuscript:**

Page 10 line 9: before the sentence "Furthermore, absorption bands ..." we added "This is likely due to the fact that  $Ca(NO_3)_2$  is more hygroscopic and soluble than  $CaCO_3$  particles."

**6. Page 12, equation 2 and 3: Why are there two formulae for the calculation of reactive uptake coefficient? One uses dN(NO2) and the other uses dNO3?**

**Reply:** Thanks for the reviewer's comments. In the equation 2 and 3,  $N(NO_2)$  is the number of reactive  $NO_2$  collisions with the surface and  $\{NO_3\}$  is surface concentrations of the nitrate.  $dN(NO_2)/dt$  represents the rate of the reactive collisions with the surface and  $d\{NO_3\}/dt$  means the nitrate formation rate. The reactive uptake coefficient ( $\gamma$ ) is defined as the rate of the reactive collisions with the surface divided by the total number of surface collisions per unit time (Z) as expressed in equation 2. In the reaction of  $NO_2$  with CaCO3 particles and  $(NH_4)_2SO_4$ -CaCO3 mixtures, the reactive  $NO_2$  collisions with the surface lead to the formation of  $NO_3^-$ . Thus the rate of the reactive  $NO_2$  collisions with the surface can be quantified in terms of the nitrate formation rate (Börensen et al., 2000; Li et al., 2006; Tong et al., 2010; Ullerstam et al., 2002).

**Related changes included in the revised manuscript:**

Page 12 line 14-15: the sentence "The rate of reactive collision can be obtained from the nitrate formation rate  $d\{NO_3^-\}/dt$ , ..." was revised to "The rate of reactive  $NO_2$ collision with the surface can be quantified in terms of the nitrate formation rate  $d\{NO_3^-\}/dt$ , ..."

**References:**

*Börensen, C., Kirchner, U., Scheer, V., Vogt, R., and Zellner, R.: Mechanism and kinetics of the reactions of*  $NO_2$  *or*  $HNO_3$  *with alumina as a mineral dust model compound, J. Phys. Chem. A, 104, 5036-5045, doi:10.1021/jp994170d, 2000.*

Li, L., Chen, Z. M., Zhang, Y. H., Zhu, T., Li, J. L., and Ding, J.: Kinetics and mechanism of heterogeneous oxidation of sulfur dioxide by ozone on surface of calcium carbonate, Atmos. Chem. Phys., 6, 2453–2464, 2006.

Tong, S. R., Wu, L. Y., Ge, M. F., Wang, W. G., and Pu, Z. F.: Heterogeneous chemistry of monocarboxylic acids on α-Al2O3 at different relative humidities, Atmos. Chem. Phys., 10, 7561-7574, doi:10.5194/acp-10-7561-2010, 2010.

Ullerstam, M., Vogt, R., Langer, S., and Ljungstrom, E.: The kinetics and mechanism of  $SO_2$  oxidation by  $O_3$  on mineral dust, Phys. Chem. Chem. Phys., 4, 4694-4699, doi:10.1039/b203529b, 2002.

7. Page 14, line 1 : They report that the amorphous hydrate Ca(NO3)2 has weak inter action with (NH4)2SO4, but the following sentence ( same page, line 5), they suggest that Ca(NO3)2 could interact with (NH4)2SO4 to form NH4NO3. How do they explain these contradictory statements?

**Reply:** Thanks for the reviewer's comments. It was reported that  $Ca(NO_3)_2$  was in the state of amorphous state at RH < 7% and solution droplets at RH > 10%. In this study, the  $Ca(NO_3)_2$  was in the state of amorphous state under dry condition and solution droplets at 40% RH (page 8 line 10-14). Also the chemical interaction of  $Ca(NO_3)_2$  with  $(NH_4)_2SO_4$  is responsible for the formation of  $CaSO_4 \ 0.5H_2O$  under dry condition and 40% RH. Figure 1a indicates that the IR absorption bands of  $CaSO_4 \ 0.5H_2O$  are weak under dry condition (the vibration modes of water group in  $CaSO_4 \ 0.5H_2O$  are too weak to be identified), while the IR peaks of  $CaSO_4 \ 0.5H_2O$  can be clearly observed at 40% RH (Figure 1b). The results indicate that the chemical interaction of  $Ca(NO_3)_2$  with  $(NH_4)_2SO_4$  is enhanced with the deliquescence of  $Ca(NO_3)_2$ . The possible reasons are that the deliquesced  $Ca(NO_3)_2$  leads to more water uptake on the mixture surfaces and that the ionic mobility of the surface ions are improved in solution droplets.

**Related changes included in the revised manuscript:**

The sentence "Meanwhile, the deliquesced  $Ca(NO_3)_2$  could interact with  $(NH_4)_2SO_4$

particles to form microcrystallites of  $NH_4NO_3$  and  $CaSO_4 nH_2O_5$ ..." on page 14 line 3-4 was revised to "Meanwhile, the chemical interaction of  $Ca(NO_3)_2$  with  $(NH_4)_2SO_4$  is enhanced with the deliquescence of  $Ca(NO_3)_2$ , resulting in the formation of microcrystallites of  $NH_4NO_3$  and  $CaSO_4 nH_2O$ ."

**Technical corrections:**

**8. Page 3 line 2: should be "gaseous", not gases.**

Reply: Page 3 line 2: "gases" was revised to "gaseous".

**9. Page 3 line 8: remove the "1" in front of "Pathak".**

Reply: Page 3 line 8: "Pathakl" was revised to "Pathak".

**10. Page 3 line 16: "... significant relevance". Incomplete sentences, relevance to what?**

**Reply:** Page 3 line 15: the sentence "… to explain the significant relevance." was revised to "… to explain these phenomena."

**11. Page 3 line 22: change "was" to "were".**

**Reply:** We revised "was" to "were" on page 3 line 20 in the sentence "Modeling studies indicated that mineral aerosols were highly associated with nitrate formation in the atmosphere."

**12. Page 3 line 26: "... after being exposed to...".**

**Reply:** We revised "... after being exposed to ..." on page 3 line 24: "Calcium carbonate particle is converted to calcium nitrate after reaction with nitrogen oxides and  $HNO_3$  in the atmosphere"

**13. Page 4 line 4: "...attributing it to...".**

**Reply:** Thanks for the reviewer's comments. We revised the sentence "... attributing

to the interaction of ions under humid condition." to "... as a result of the interaction of ions under humid condition."

**14. Page 4 line 12: what do they mean by, "The catalysis and basic coexists could ..."**

**Reply:** Thanks for the reviewer's comments. The original sentence in the paper of Li et al. is "The catalytic and basic additives could enhance the production of sulfate on the NaCl surface." Their results showed that the additive of basic additives (e.g. MgO and CaCO3) could greatly increase the basic property of the surface of NaCl and that  $SO_2$  could easily absorbe on the alkaline surface and subsequently be oxidized into sulfate by  $O_3$ .

**Related changes included in the revised manuscript:**

The sentence "The catalysis and basic coexists could increase the formation of sulfate on NaCl particle surfaces." on page 4 line 10-11 was revised to "The catalytic and basic additives, e.g., MgO and CaCO3, could increase the basic property of the surface of NaCl and increase the formation of sulfate by facilitating the absorbance of  $SO_2$  on the alkaline surface."

**15. Page 6 line 9: add "respectively" at the end of the sentence.**

**Reply:** We are thankful for this comment. We had added "respectively" at the end of the sentence on page 6 line 5-8 in the ACPD version.

16. Page 7 line 7: How "dry" (< 1% humidity?) were the experimental conditions? It has been reported in literature that there are enough water layers at RH <5% RH to influence surface reactions. Their "dry" experimental (RH) conditions should be presented.

**Reply:** Thanks for the reviewer's comment. Dry condition represents an experiment condition that the gases entering reactor chamber with a very low relative humidity (*RH*). In this study, the condition when the gases are dehumidified by silica gel and

molecular sieve to less than 1% RH before flowing into DRIFTs reactor chamber is called dry condition (page 5 line 28; page 6 line 1). The RH and temperature of the inflow of sample cell are measured using a commercial humidity and temperature sensor (HMT330; Vaisala) with a measurement accuracy of  $\pm 1\%$  RH and  $\pm 0.2^{\circ}C$ , respectively (page 6, line 6-8).

"Dry condition", actually, is widely used to describe a very low RH experiment condition in scientific papers. Goodman et al. (2000, 2001) used "dry condition" and "conditions near 0 relative humidity" in their papers. Al-Abadleh et al. (2004) described their experiment conditions as "under dry (<1% RH) and wet (20-25%) conditions". Also, Al-Hosney et al. (2005) described the condition as "under dry conditions near 0% RH", Li et al. (2010) described it as "dry condition (RH<10%)", and Tong et al. (2010) used "dry condition RH<1%".

**References:**

Al-Abadleh, H. A., Al-Hosney, H. A., and Grassian, V. H.: Oxide and carbonate surfaces as environmental interfaces: the importance of water in surface composition and surface reactivity, J. Mol. Catal. A: Chem., 228, 47-54, doi:10.1016/j.molcata.2004.09.059, 2004.

Al-Hosney, H. A., and Grassian, V. H.: Water, sulfur dioxide and nitric acid adsorption on calcium carbonate: A transmission and ATR-FTIR study, Phys. Chem. Chem. Phys., 7, 1266-1276, doi:10.1039/b417872f, 2005.

Goodman, A. L., Bernard, E. T., and Grassian, V. H.: Spectroscopic study of nitric acid and water adsorption on oxide particles: Enhanced nitric acid uptake kinetics in the presence of adsorbed water, J. Phys. Chem. A, 105, 6443-6457, doi:10.1021/jp0037221, 2001.

Goodman, A. L., Underwood, G. M., and Grassian, V. H.: A laboratory study of the heterogeneous reaction of nitric acid on calcium carbonate particles, J. Geophys. Res., 105, 29053-29064, doi:10.1029/2000jd900396, 2000.

Li, H. J., Zhu, T., Zhao, D. F., Zhang, Z. F., and Chen, Z. M. : Kinetics and mechanisms of heterogeneous reaction of NO2 on CaCO3 surfaces under dry and wet conditions, Atmos. Chem. Phys., 10, 463–474, 2010.

Tong, S. R., Wu, L. Y., Ge, M. F., Wang, W. G., and Pu, Z. F.: Heterogeneous chemistry of monocarboxylic acids on  $\alpha$ -Al2O3 at different relative humidities, Atmos. Chem. Phys., 10, 7561-7574, doi:10.5194/acp-10-7561-2010, 2010. Ullerstam, M., Vogt, R., Langer, S., and Ljungstrom, E.: The kinetics and mechanism of SO2 oxidation by O3 on mineral dust, Phys. Chem. Chem. Phys., 4, 4694-4699, doi:10.1039/b203529b, 2002.

**17. Page 9 line 23: "...can be concluded..."**

**Reply:** Thanks for the reviewer's comment. We revised the sentence "In addition, it can be concluded from Fig. 1 that  $NO_2$  did not show..." on page 9 line 25 to "In conclusion,  $NO_2$  did not show..."

**18. Page 10 line 1: "...add a comma after N2, its confusing without it.**

**Reply:** Thanks for the reviewer's comment. We revised the sentence "In another set of experiments, the mixture of FAS-57 was exposed to nitrogen with corresponding RHs to investigate ..." on page 10 line 1-2 to "In another set of experiments, the mixture of FAS-57 was exposed to nitrogen, with corresponding RHs in order to investigate ...".

**19.** Page 11, line 17: where are the "... stable formation" states/rates? This statement needs to be explained.**

**Reply:** We are thankful for the reviewer's comment. Figure 3a represents the integrated absorbance of nitrate as a function of time under dry condition. It suggests that the formation of nitrate on  $CaCO_3$  and the  $CaCO_3$ - $(NH_4)_2SO_4$  mixtures surfaces can be divided into three stages. Stage 1: the integrated absorbance of nitrate increases linearly with time and it is called initial stage. In this stage, the nitrate formation rate on particle surfaces is faster than in the other two stages. Stage 2 (transition stage): the increase of the integrated absorbance of nitrate slowed down. Stage 3: the integrated absorbance of nitrate increase at a relatively stable rate that is much smaller than that at stage 1 (Wu et al., 2013; Li et al., 2006; Li et al. 2010)

**Related changes included in the revised manuscript:**

We revised the sentence "The nitrate formation rates were fast at initial stage and then slowed down after a transition stage under dry condition." on page 10 line 25-26 to "The formation of nitrate on sample surfaces could be divided into three stages under dry conditions. The integrated absorbance of nitrate increased linearly with time in initial stage and it finally increase at a stable rate after a transition period."

**References:**

Li, H. J., Zhu, T., Zhao, D. F., Zhang, Z. F., and Chen, Z. M. : Kinetics and mechanisms of heterogeneous reaction of NO2 on CaCO3 surfaces under dry and wet conditions, Atmos. Chem. Phys., 10, 463–474, 2010.

Li, L., Chen, Z. M., Zhang, Y. H., Zhu, T., Li, J. L., and Ding, J.: Kinetics and mechanism of heterogeneous oxidation of sulfur dioxide by ozone on surface of calcium carbonate, Atmos. Chem. Phys., 6, 2453–2464, 2006.

Wu, L. Y., Tong, S. R., and Ge, M. F.: Heterogeneous reaction of NO2 on Al2O3: the effect of temperature on the nitrite and nitrate formation, J. Phys. Chem. A, 117, 4937-4944, doi:10.1021/jp402773c, 2013.

**20. Page 12, line 10: create a better notation for the effective surface area, because "As" is confusing.**

**Reply:** We appreciate for the reviewer's comment. We agree that "As" is a little confusing in the sentence "Where  $N(NO_2)$  is the number of reactive  $NO_2$  collisions with the surface, As is the effective surface area of samples and  $[NO_2]$  is the gas-phase concentration of  $NO_2$ " on page 12 line 11-12. We revised "As" to "Asurface" to avoid confusion (Ullerstam et al., 2002).

**References:**

Ullerstam, M., Vogt, R., Langer, S., and Ljungstrom, E.: The kinetics and mechanism of  $SO_2$  oxidation by  $O_3$  on mineral dust, Phys. Chem. Chem. Phys., 4, 4694-4699, doi:10.1039/b203529b, 2002.

**21. Page 13, line 29: remove "with absence of water vapor". It's redundant since**

**you have mentioned "under dry conditions" at the beginning of the sentence.**

**Reply:** We deleted "with the absence of water vapor" in the sentence "Under dry condition, little reaction occurs between  $CaCO_3$  and  $(NH_4)_2SO_4$ ." on page 13 line 27-28.

**22. Page 14, line 9: "...decreased..."**

**Reply:** We had revised "decrease" to "decreased" in the sentence "However, the nitrate formation rates and nitrate concentrations at 60% RH decreased compared to those at 40% RH for the mixtures with mass percentage of  $(NH_4)_2SO_4$  larger than 57%." on page 14, line 7-9.

---

## Author Response (AR2)

**Response**

**Response to referee #1:**

We are grateful to Referee #1 for the comments and the constructive suggestions to improve our manuscript. Our response to the comments and changes to the manuscript are included below. We repeat the specific points raised by the reviewer in bold font, followed by our response in italic font. The pages numbers and lines mentioned below are consistent with those in the Atmospheric Chemistry and Physics Discussions (ACPD) paper.

**Mineral dust and sulfate are common components in atmospheric particulate maters (PMs), and their coagulation in the atmosphere can form new types of PMs whose physical and chemical characters will be altered, thus affecting on atmospheric physical and chemical processes. Because only few studies investigated the heterogeneous reactions under complex conditions, there is still large gap to explain many phenomena of field measurements by using the current knowledge of atmospheric chemistry. The new finding in this study about the heterogeneous reactions of $NO_2$ on the surface of $CaCO_3$-$(NH_4)_2SO_4$ mixtures provided important information, that is, the heterogeneous reactions in the atmosphere may play important role on formation of nitrate, $CaSO_4 \ 0.5H_2O$ (bassanite), $CaSO_4 \ 2H_2O$ (gypsum) and $(NH_4)_2Ca(SO_4)_2 \ H_2O$ (koktaite). This reviewer recommends the manuscript be published in the journal after considering the following one specific: The formation rates of nitrate based on the integrated absorbance of the IR peak area between 1390 and 1250 cm -1 are inconsistent with the final concentrations of nitrate measured by IC. The intensity of the IR absorbance in the DRIFTS can only reveal the surface concentration of nitrate, whereas the nitrate concentrations measured by IC are the bulk concentrations in the PMs. The surface nitrate formed through the heterogeneous reactions was suspected to easily diffuse into inner layer of the PMs under high RH conditions. The authors should present the explanations.**

*Reply: Thanks for the reviewer's comment. We agree with the reviewer's point that the*

*infrared beam primarily interrogates the upper portion of the particulate samples and IR absorbance in the DRIFTS reveals the surface concentration of nitrate. The infrared beam is, however, not restricted solely to the top few particle layers (Vogt and Finlaysonpitts, 1994). In fact, DRIFTS has been applied extensively to study the kinetics and mechanisms of gas-solid reactions (Ullerstam et al., 2002, 2003; Li et al., 2007; Li et al., 2010; Tong et al. 2010), where the kinetics of the reactions are followed using the integrated absorbance-reaction time behavior and the reactive uptake coefficient (γ) is determined from the infrared absorbance, that calibrated by ion chromatography (IC).*

*Vogt et al. (1994) investigated the depth of the pellet from which the infrared signal generated. They mixed homogeneous $NaNO_3$ or NaCl mixture with an additional 0.2-0.5mm layers of neat NaCl powder on top and found that significant signals could still be observed. Also they reported that the absorbance integrated over the $v_3$ region was linearly dependent on the amount of nitrate determined by IC in the reaction of solid NaCl with gaseous $NO_2$ and $HNO_3$. In this study, as indicated in Figure S3 that over a large concentration range the integrated nitrate absorbance over the $v_3$ region (1390 to 1250 $cm^{-1}$) was proportional to the nitrate ions concentration detected by IC. In another word, the reacted particles were within the depth that infrared signal generated and the nitrate formed during the reaction could be presented by the integrated absorbance over the $v_3$ region.*

*Li et al. (2010) investigated the heterogeneous reaction of $CaCO_3$ with $NO_2$ using a DRIFTS reactor and found that there was a linear relationship between absorbance integrated over the $v_1$ region (1013-1073 $cm^{-1}$) and the number of nitrate ions determined by IC. They reported that the conversion factor f (in the equation of the nitrate ions $\{NO_3^-\}$ = (integrated absorbance $I_A$) × f) was found to be independent of reaction time and $NO_2$ concentration as long as the experiment was completed at a stage when the absorption of the nitrate band was still growing. In this research, we found that the integrated nitrate absorbance over the $v_1$ region (1013-1073 $cm^{-1}$) and the $v_3$ region (1250-1390 $cm^{-1}$) could well overlap after the former multiplied by a constant (see in Figure S2). And all the experiments in our study were completed as the integrated nitrate absorptions were still growing (Figure 3).*

*Therefore the nitrate formation rates based on the integrated absorbance of the IR peak area*

*between 1390-1250 were well consistent with the nitrate concentrations measured by IC.*

*Following the referee's suggestion, we added a series of experiments that stopped the heterogeneous reactions of $NO_2$ with $CaCO_3$ and $CaCO_3$-$(NH_4)_2SO_4$ mixtures at the initial stage (e.g. 20min, 30min, and 40min) under wet and dry conditions. The results indicated that the integrated absorbance of nitrate has a linear relationship with the amount of nitrate determined by ion chromatography during all the reaction periods. Some modifications have been made in Figure S3 after some data added to the calibration plot. Figure S2 was added to illustrate that the integrated nitrate absorbance over the $v_1$ region (1013-1073 $cm^{-1}$) and the $v_3$ region (1250-1390 $cm^{-1}$) could well overlap after the former multiplied by a constant.*

**Related changes included in the revised manuscript:**

*Page 10 line 24, after the sentence "… interruption of the absorptions of sulfates" we added "The integrated nitrate absorbance over the $v_1$ region (1013-1073 $cm^{-1}$) and the $v_3$ region (1250-1390 $cm^{-1}$) could well overlap after the former multiplied by a constant (Figure S2)."*

*Page 11 line 12, we revised "Fig. S2" in the sentence "… at 60%, 40% RH and dry condition (see Fig. S2)." to Fig S3.*

[Figure]

**Figure S2:** *The integrated nitrate absorbance over the $v_1$ region (1013-1073 $cm^{-1}$) and the $v_3$ region (1250-1390 $cm^{-1}$) after the $v_1$ region multiplied by 3, 7, 10, 12 under dry condition, 40% RH, 60% RH, and 85% RH, respectively, for the reaction of $NO_2$ with $CaCO_3$ particles.*

[Figure]

**Figure S3**. *The number of $NO_3^-$ ions detected by IC as a function of integrated absorbance of IR peak between 1390 and 1250 $cm^{-1}$ at (a) dry condition, 40% RH, and 60% RH, (b) 85% RH.*

***References:***

*Li, H. J., Zhu, T., Zhao, D. F., Zhang, Z. F., and Chen, Z. M. : Kinetics and mechanisms of heterogeneous reaction of $NO_2$ on $CaCO_3$ surfaces under dry and wet conditions, Atmos. Chem. Phys., 10, 463–474, 2010.*

15 *Li, L., Chen, Z. M., Zhang, Y. H., Zhu, T., Li, S., Li, H. J., Zhu, L. H., and Xu, B. Y.: Heterogeneous oxidation of sulfur dioxide by ozone on the surface of sodium chloride and its mixtures with other components, J. Geophys. Res.-Atmos., 112,*

*doi:10.1029/2006jd008207, 2007.*

*Li, L., Chen, Z. M., Zhang, Y. H., Zhu, T., Li, J. L., and Ding, J.: Kinetics and mechanism of heterogeneous oxidation of sulfur dioxide by ozone on surface of calcium carbonate, Atmos. Chem. Phys., 6, 2453–2464, 2006.*

*Liu, Y. J., Zhu, T., Zhao, D. F., and Zhang, Z. F.: Investigation of the hygroscopic properties of $Ca(NO_3)_2$ and internally mixed $Ca(NO_3)_2/CaCO_3$ particles by micro-Raman spectrometry, Atmos. Chem. Phys., 8, 7205–7215, 2008.*

*Tang, I. N., and Fung, K. H.: Hydration and Raman scattering studies of levitated microparticles: $Ba(NO_3)_2$, $Sr(NO_3)_2$, and $Ca(NO_3)_2$, J. Chem. Phys., 106, 1653-1660, doi:10.1063/1.473318, 1997.*

*Tong, S. R., Wu, L. Y., Ge, M. F., Wang, W. G., and Pu, Z. F.: Heterogeneous chemistry of monocarboxylic acids on $\alpha$-$Al_2O_3$ at different relative humidities, Atmos. Chem. Phys., 10, 7561-7574, doi:10.5194/acp-10-7561-2010, 2010.*

*Ullerstam, M., Vogt, R., Langer, S., and Ljungstrom, E.: The kinetics and mechanism of $SO_2$ oxidation by $O_3$ on mineral dust, Phys. Chem. Chem. Phys., 4, 4694-4699, doi:10.1039/b203529b, 2002.*

*Ullerstam, M., Johnson, M. S., Vogt, R., and Ljungstrom, E.: DRIFTS and Knudsen cell study of the heterogeneous reactivity of $SO_2$ and $NO_2$ on mineral dust, Atmos. Chem. Phys., 3, 2043-2051, 2003.*

*Vogt, R., and Finlaysonpitts, B. J.: A Diffuse Reflectance Infrared Fourier-Transform Spectroscopic (DRIFTS) study of the surface-reaction of NaCl with gaseous $NO_2$ and $HNO_3$, J. Phys. Chem., 98, 3747-3755, doi:10.1021/j100065a033, 1994.*

*Wu, L. Y., Tong, S. R., and Ge, M. F.: Heterogeneous reaction of $NO_2$ on $Al_2O_3$: the effect of temperature on the nitrite and nitrate formation, J. Phys. Chem. A, 117, 4937-4944, doi:10.1021/jp402773c, 2013.*

**Response to referee #2:**

We are grateful to Referee #2 for giving valuable comments and helpful suggestions to improve our manuscript. Our response to the comments and changes to the manuscript are included below. We repeat the specific points raised by the reviewer in bold font, followed by our response in italic font. The manuscript that referee #2 commented is the version that delivered to Atmospheric Chemistry and Physics (ACP) initially, while we had made some small modifications according to the suggestions from Referee #1 in the ACPD version. The numbers of pages and lines are consistent with those in the ACPD paper to avoid misunderstanding.

**General comments: The article aims to understand the uptake and kinetic behavior of a mixed aerosols system with its reaction with $NO_2$. The article has laid out all the aspects of the experiments and presented the data well. The role of $(NH_4)_2SO_4$ in the reaction was analyzed well. The data, summaries and mechanisms fits well but there are few major contradictory statements made in the different sections of the article that need clarification.**

**I recommend publication after a rewrite clarifying some of the major contradictory statement highlighted below:**

**Reply:** *We appreciate the reviewer's comments. And we have carefully revised our manuscript according to the reviewer's suggestions.*

**Specific comments**

**1. The main issue I have is the role of $(NH_4)_2SO_4$ in the reaction. There seems to be two contradictory summaries being presented here, without explanation on how/why the $(NH_4)_2SO_4$ is causing these effects. There seems to be a cutoff RH value (60%), below which the effect of $(NH_4)_2SO_4$ is promotive and above which the effect is opposite (see page 10, line 26; page 12, line 3; page 13, line 19, or 21 ;). The authors have proposed active site dependence, (page 10, line 26) and deliquescence of $(NH_4)_2SO_4$ (page 14, line 16) as possible reasons for this. The way the sample mixture was made (page 5**

**line 21) contradicts the first reason; and these negative effect starts at 60% RH (which is further lower that DRH of (NH₄)₂SO₄ contradicts the second reason. The role of (NH₄)₂SO₄ is important (as the authors have clearly shown), their reasons for the observed effects need more explanations, and these contradictory statements do not help the reader/article.**

*Reply: Thanks for the reviewer's comment. We were regretful that we did not clarify enough about how $(NH_4)_2SO_4$ was causing effects in the heterogeneous reaction, resulting in misunderstanding of the reviewer and reader. In fact, we did not mention that active site dependence and the deliquescence of $(NH_4)_2SO_4$ were responsible for the effects of $(NH_4)_2SO_4$ in this paper. And 60% RH is not a cutoff RH between the two opposite effects.* **We did emphasize that the chemical interaction of $(NH_4)_2SO_4$ with $Ca(NO_3)_2$ or $CaCO_3$ were the possible reasons for the promotive or inhibiting effects (page 10 line 1-19, page 13 line 12-14, page 13 line 27-29, page 14 line 1-15) and the nitrate concentrations were enhanced under all the wet conditions investigated (40%, 60% and 85% RH) (page 13 line 14-21 and line 23-25).**

*Firstly, $(NH_4)_2SO_4$ has little effects on nitrate formation in the heterogeneous reaction of the mixtures with $NO_2$ under dry condition (page 13 line 27-29 and page 14 line 1). Figure 1a indicated that $(NH_4)_2SO_4$ particles has limited interaction with the amorphous state $Ca(NO_3)_2$ and Figure 2 suggested that it has little reaction with $CaCO_3$ particles under dry condition. The reactive sites dependence was the possible reason to explain the results that the lasting time of initial stages and the $NO_3^-$ mass concentrations decrease linearly with increasing $(NH_4)_2SO_4$ content in the mixtures (as the reviewer mentioned, on page 10 line 27-28 in initial manuscript version), since the nitrate is produced from the uptake of $NO_2$ on $CaCO_3$ particles without the participation of $(NH_4)_2SO_4$ under dry condition.*

*As RH increased from dry condition to* **40% RH, the chemical reaction of $CaCO_3$ with $(NH_4)_2SO_4$ particles is still neglectable (Figure 2). And the chemical interaction of the deliquesced $Ca(NO_3)_2$ with $(NH_4)_2SO_4$ particles are responsible for the formation of $NH_4NO_3$ and $CaSO_4 0.5H_2O$**, *which may enhance the ionic mobility of the surface ions (Allen et al., 1996), modify the surface structure and re-expose reactive sites (Al-Hosney and*

*Grassian, 2005), consequently showing promotive effects on the nitrate formation during the heterogeneous reaction of $NO_2$ with the mixtures.*

*At 60% RH, **a chemical reaction in the coagulation of $CaCO_3$ and $(NH_4)_2SO_4$ particles actually occurs without the introduction of $NO_2$** (page 10 line 9-11). Consequently, $CaCO_3$ particles are partly consumed during the coagulation with $(NH_4)_2SO_4$ and the $CaSO_4\, nH_2O$ formed in the coagulation may block reactive sites for further reaction, resulting in an inhibiting effect on nitrate formation (page 14 line 9-15). At the same time, the deliquesced $Ca(NO_3)_2$ still has chemical interactions with $(NH_4)_2SO_4$ (page 10 line 13-19). **Therefore, there is a combined effect of the two opposing effects from the interaction of $(NH_4)_2SO_4$ with $Ca(NO_3)_2$ and the interaction of $(NH_4)_2SO_4$ with $CaCO_3$.** Furthermore, it is well consistent with the results that the nitrate formation rates and the $NO_3^-$ mass concentrations at 60% RH are slightly larger than those at 40% RH for the mixtures with mass percentage of $(NH_4)_2SO_4$ smaller than 43%, while it is opposite for the mixtures with mass percentage of $(NH_4)_2SO_4$ larger than 57%. **Thus 60% RH is not a cutoff value.***

*As for 85% RH, **the deliquescence of $(NH_4)_2SO_4$ (Cziczo et al., 1997) leads to more water uptake on the mixture surfaces, facilitating the reaction of $(NH_4)_2SO_4$ with $CaCO_3$ (page 14 line15-16).** Therefore the negative effects are more obvious at 85% RH than at 60% and 40% RH. It should be noticed that although the nitrate formation rates and $NO_3^-$ mass concentrations at 85% RH are smaller than those at 60% and 40% RH, the nitrate concentrations are still improved at 85% RH (page 13 line 17-25).*

*Some modifications have been made in order to clarify clearly how $(NH_4)_2SO_4$ affects the nitrate formation in the heterogeneous reaction of $NO_2$ with $CaCO_3$-$(NH_4)_2SO_4$ mixtures at different RHs.*

***Related changes included in the revised manuscript:***

*Page 2 line 15-17: the sentence "Under wet conditions, the $CaCO_3$-$(NH_4)_2SO_4$ mixtures exhibited…." was revised to "Under wet conditions, the chemical interaction of $(NH_4)_2SO_4$ with $Ca(NO_3)_2$ has a promotive effects on the nitrate formation in the heterogeneous reaction of the mixtures with $NO_2$, while the coagulation of $(NH_4)_2SO_4$ with $CaCO_3$ exhibits an inhibiting effects at the same time. The nitrate formation is promoted in the heterogeneous*

*reaction of NO$_2$ with CaCO$_3$-(NH$_4$)$_2$SO$_4$ mixtures, especially at medium RHs."*

*Page 10 line 7-9: the sentence "… therefore the heterogeneous reactions of NO$_2$ with the CaCO$_3$-(NH$_4$)$_2$SO$_4$ mixtures were responsible for the formation of bassanite." was revised to "… therefore the chemical interaction of Ca(NO$_3$)$_2$ with (NH$_4$)$_2$SO$_4$ was responsible for the*
5    *formation of bassanite in these conditions."*

*Page 10 line 17-19: the sentence "And there were additional gypsum and koktaite products formed…." was revised to "Thus CaSO$_4$ nH$_2$O and koktaite products could be formed both from the chemical interaction of (NH$_4$)$_2$SO$_4$ with Ca(NO$_3$)$_2$ and the reaction of (NH$_4$)$_2$SO$_4$ with CaCO$_3$ at 60% and 85% RH."*

10    *Page 11 line 22, after the sentence "…followed by that at 85% RH" we added "While for the mixtures with mass fraction of (NH$_4$)$_2$SO$_4$ smaller than 43%, the nitrate formation rates increased initially as RH elevated from 40% RH to 60% RH then it decreased obviously as RH increased to 85% RH. The differences in the tend of the nitrate formation rates with RH for the mixtures could be explained by the combined opposite effects from the interaction of*
15    *(NH$_4$)$_2$SO$_4$ with Ca(NO$_3$)$_2$ or CaCO$_3$ at 60% RH."*

*Page 11 line 26, after the sentence "…at corresponding RH, respectively" we added "As RH increased from dry condition to 40% and 60% RH, the initial nitrate formation rates decreased less for the reaction of NO$_2$ with the mixtures than with CaCO$_3$ particles, while it was opposite as RH increased to 85% RH"*

20    *Page 13 line 17-18: before the sentence "The NO$_3^-$ mass concentrations for the mixture of FAS-57…" we added "The NO$_3^-$ mass concentrations increase much more for the mixtures than for pure CaCO$_3$ particles as RH elevated from dry condition to wet conditions, e.g."*

*Page 14 line 7: after the sentence "…expose additional active sites on CaCO$_3$ particles in the mixtures" we added "Thus the chemical interaction of Ca(NO$_3$)$_2$ and (NH$_4$)$_2$SO$_4$ particles*
25    *may exhibits promotive effects on the nitrate formation during the heterogeneous reaction of NO$_2$ with CaCO$_3$-(NH$_4$)$_2$SO$_4$ mixtures."*

*Page 14 line 7-9: the sentence "However, the nitrate formation rates and nitrate concentrations at 60% RH was decreased compared to those at 40% RH for the mixtures with mass percentage of (NH$_4$)$_2$SO$_4$ larger than 57%." was revised to "The nitrate formation rates*

*and nitrate concentrations increase slightly when RH increased from 40% RH to 60% RH for the mixtures with mass percentage of $(NH_4)_2SO_4$ less than 43%. However, it was opposite for the mixtures with mass percentage of $(NH_4)_2SO_4$ larger than 57% that the nitrate formation rates and nitrate concentrations at 60% RH are smaller than those at 40% RH."*

5     *References:*

*Al-Abadleh, H. A., Al-Hosney, H. A., and Grassian, V. H.: Oxide and carbonate surfaces as environmental interfaces: the importance of water in surface composition and surface reactivity, J. Mol. Catal. A: Chem., 228, 47-54, doi:10.1016/j.molcata.2004.09.059, 2004.*

10  *Al-Hosney, H. A., and Grassian, V. H.: Water, sulfur dioxide and nitric acid adsorption on calcium carbonate: A transmission and ATR-FTIR study, Phys. Chem. Chem. Phys., 7, 1266-1276, doi:10.1039/b417872f, 2005.*

*Allen, H. C., Laux, J. M., Vogt, R., Finlayson-Pitts, B. J., and Hemminger, J. C.: Water-induced reorganization of ultrathin nitrate films on NaCl: Implications for the*

15  *tropospheric chemistry of sea salt particles, J. Phys. Chem., 100, 6371-6375, doi:10.1021/jp953675a, 1996.*

*Mori, I., Nishikawa, M., and Iwasaka, Y.: Chemical reaction during the coagulation of ammonium sulphate and mineral particles in the atmosphere, Sci. Tot. Environ., 224, 87-91, doi:10.1016/s0048-9697(98)00323-4, 1998.*

**2. Page 9, line 2-5: The identification of $CaSO_4.0.5H_2O$ and $CaSO_4.2H_2O$ uses very similar IR peaks. It's not entirely clear how these same peaks were used to differentiate the $CaSO_4.0.5H_2O$ from the $CaSO_4.2H_2O$.**

**Reply:** *Thanks for the reviewer's comment. The IR absorption peaks at 1008 and 1116 $cm^{-1}$*

25  *due to $CaSO_4.0.5H_2O$ and the peaks at 1005 and 1117 $cm^{-1}$ due to $CaSO_4.2H_2O$ are hard to distinguish. There are, actually, some features that can be used to differentiate $CaSO_4.0.5H_2O$ from $CaSO_4.2H_2O$. As has been described in this paper (page 9, line 6-9), the peaks at 1096 and 1155 $cm^{-1}$ belong to $CaSO_4.0.5H_2O$ can be clearly observed in the IR spectrum, which are evidences for the formation of $CaSO_4.0.5H_2O$ rather than $CaSO_4.2H_2O$. Besides,*

*CaSO$_4$.2H$_2$O shows two IR-active modes in the bending modes of crystal hydrate water at 1620 and 1685 cm$^{-1}$, while CaSO$_4$.0.5H$_2$O has only one band at 1620 cm$^{-1}$. Furthermore, the two stretching modes of crystal hydrate water occur at 3495, 3545 and 3400 cm$^{-1}$ for CaSO$_4$.2H$_2$O, at 3555 and 3610 cm$^{-1}$ for CaSO$_4$.0.5H$_2$O (Prasad, 2005; Liu et al., 2009).*

5      ***Related changes included in the revised manuscript:***

*Page 9 line 4-6: the sentence "The IR absorption bands of ..." was revised to "Although the IR absorption bands of bassanite and gypsum had some overlaps in the region between 1000 and 1250 cm$^{-1}$, there were some features that could be used to differentiate CaSO$_4$.0.5H$_2$O from CaSO$_4$.2H$_2$O."*

10      *References:*

*Liu, Y., Wang, A., Freeman, J. J.: Raman, Mir, and NIR spectroscopic study of calcium sulfates: gypsum,bassanite, and anhydrite, 40th Lunar and Planetary Science Conference, 2009.*

*Prasad, P. S. R., Krishna Chaitanya, V., Shiva Prasad, K., and Narayana Rao, D.: Direct*
15      *formation of the γ-CaSO$_4$ phase in dehydration process of gypsum: In situ FTIR study, Am. Mineral., 90, 672-678, doi:10.2138/am.2005.1742, 2005.*

**3. Page 9 line 21: How is the decomposition of CaCO3 manifest itself as an increasing intensity of the 1570 cm$^{-1}$ band? Decomposition usually leads to a negative (loss of) intensity, not a positive**
20      **(increasing) intensity. The 1570 cm$^{-1}$ has been assigned to HSO$_4^-$, how is the increasing intensity of this peak tie-in to the loss of CaCO$_3$? I am assuming it's from a specific reaction, but this is not clearly stated here.**

**Reply:** *Thanks for the reviewer's suggestions. Normally, the decomposition of reactants leads to a negative intensity of IR spectrum in DRIFTS experiments.* **In this study, the IR**
25      **absorption peak at 1570 cm$^{-1}$ is assigned to the asymmetric stretching of HCO$_3^-$** *(Al-Hosney et al., 2004; Li et al., 2010). In fact, there is no interruption from the IR absorption bands of other reactants and products in this range.* **The positive intensity is likely due to the increasing information of HCO$_3^-$, which is from the decomposition of bulk CaCO$_3$ under wet conditions.** *As indicated in Figure 1, the peak at 1570 cm$^{-1}$ did not appear under dry*

*condition and it increased with increasing RH. The reactions are limited to surfaces and $H_2CO_3$ can exist as absorbed phase under dry condition. While the reaction of $NO_2$ can occur not only on the surfaces of $CaCO_3$ and mixtures but also into the bulk of the samples in the presence of surface condensed water (Goodman et al., 2001; Goodman et al., 2000). Furthermore, the acidity of surface condensed water is enhanced as a result of the formation of $HNO_3$ and the dissolution of $(NH_4)_2SO_4$, which facilitates the decomposition of the bulk $CaCO_3$ particles.*

***Related changes included in the revised manuscript:***

*Page 9 line 23-24: the sentence "Additionally, the increasing intensity of absorption bands at $1570\ cm^{-1}$ implied that the decomposition of $CaCO_3$ was enhanced at 85% RH." was revised to "Additionally, the IR absorption peaks at $1570\ cm^{-1}$ in Figure 1d were much stronger than those at 40% and 60% RH. The positive intensity was likely due to the increasing information of $HCO_3^-$, which was from the decomposition of the bulk $CaCO_3$ under wet conditions. It could be interpreted that the reaction of $NO_2$ can occur not only on the surfaces of $CaCO_3$ and the mixtures but also into the bulk of the samples under wet conditions. Also the acidity of surface condensed water was enhanced as a result of the formation of $HNO_3$ and the dissolution of $(NH_4)_2SO_4$, which facilitates the decomposition of bulk $CaCO_3$ particles."*

***References:***

*Al-Hosney, H. A., and Grassian, V. H.: Water, sulfur dioxide and nitric acid adsorption on calcium carbonate: A transmission and ATR-FTIR study, Phys. Chem. Chem. Phys., 7, 1266-1276, doi:10.1039/b417872f, 2005.*

*Al-Hosney, H. A., and Grassian, V. H.: Carbonic Acid: an important intermediate in the surface chemistry of calcium carbonate, J. Am. Chem. Soc., 126, 8068-8069, doi:10.1021/ja0490774, 2004.*

*Goodman, A. L., Bernard, E. T., and Grassian, V. H.: Spectroscopic study of nitric acid and water adsorption on oxide particles: Enhanced nitric acid uptake kinetics in the presence of adsorbed water, J. Phys. Chem. A, 105, 6443-6457, doi:10.1021/jp003722l, 2001.*

*Goodman, A. L., Underwood, G. M., and Grassian, V. H.: A laboratory study of the heterogeneous reaction of nitric acid on calcium carbonate particles, J. Geophys. Res., 105,*

*29053-29064, doi:10.1029/2000jd900396, 2000.*

*Li, H. J., Zhu, T., Zhao, D. F., Zhang, Z. F., and Chen, Z. M. : Kinetics and mechanisms of heterogeneous reaction of $NO_2$ on $CaCO_3$ surfaces under dry and wet conditions, Atmos. Chem. Phys., 10, 463–474, 2010.*

**4. Page 9 line 27: "…surface nitrate was decreased with increased $Ca(NO_3)_2$ content..". The sentence seems contradictory, how was the surface nitrate and bulk nitrate differentiated from the spectra?**

*Reply: Thanks for the reviewer's advice. This sentence should be corrected to "Moreover, the*
10 *surface nitrate was decreased with increasing $(NH_4)_2SO_4$ content in mixtures." (page 9 line 27-28). This sentence was in the initial manuscript and it had been deleted in the ACPD version. I think it cannot differentiate surface nitrate from bulk nitrate according to IR spectra.*

15 **5. Page 10 line 14-15: "…was faster than the reaction of…" how was this (fast reaction) determined? Needs more explanation.**

*Reply: Thanks for the reviewer's advice. We realized that the sentence "This is likely due to the fact that the reaction between $(NH_4)_2SO_4$ and $Ca(NO_3)_2$ was faster than the reaction of $(NH_4)_2SO_4$ with $CaCO_3$." was misleading. What we wanted to express was that $Ca(NO_3)_2$ were more*
20 *hygroscopic and soluble than $CaCO_3$ and it may has stronger chemical interaction with $(NH_4)_2SO_4$ than $CaCO_3$ particles under the same condition. This sentence was in the initial manuscript and it had been deleted in the ACPD version.*

*Related changes included in the revised manuscript:*

*Page 10 line 9: before the sentence "Furthermore, absorption bands …" we added "This is*
25 *likely due to the fact that $Ca(NO_3)_2$ is more hygroscopic and soluble than $CaCO_3$ particles."*

**6. Page 12, equation 2 and 3: Why are there two formulae for the calculation of reactive uptake coefficient? One uses dN(NO2) and the other uses dNO3?**

*Reply: Thanks for the reviewer's comments. In the equation 2 and 3, $N(NO_2)$ is the number of*

*reactive $NO_2$ collisions with the surface and $\{NO_3\}$ is surface concentrations of the nitrate. $dN(NO_2)/dt$ represents the rate of the reactive collisions with the surface and $d\{NO_3\}/dt$ means the nitrate formation rate. The reactive uptake coefficient ($\gamma$) is defined as the rate of the reactive collisions with the surface divided by the total number of surface collisions per unit time (Z) as expressed in equation 2. In the reaction of $NO_2$ with $CaCO_3$ particles and $(NH_4)_2SO_4$-$CaCO_3$ mixtures, the reactive $NO_2$ collisions with the surface lead to the formation of $NO_3^-$. Thus the rate of the reactive $NO_2$ collisions with the surface can be quantified in terms of the nitrate formation rate (Börensen et al., 2000; Li et al., 2006; Tong et al., 2010; Ullerstam et al., 2002).*

*Related changes included in the revised manuscript:*

*Page 12 line 14-15: the sentence "The rate of reactive collision can be obtained from the nitrate formation rate $d\{NO_3^-\}/dt$, ..." was revised to "The rate of reactive $NO_2$ collision with the surface can be quantified in terms of the nitrate formation rate $d\{NO_3^-\}/dt$, ..."*

[revised manuscript text omitted]
 had some overlaps in the region between 1000 and 1250 $cm^{-1}$, there were some features that could be used to differentiate $CaSO_4.0.5H_2O$ from $CaSO_4.2H_2O$. Gypsum showed two IR-active modes in the bending modes of crystal hydrate water at 1620 and 1685 $cm^{-1}$, while bassanite had only one band at 1620 $cm^{-1}$. And the two stretching modes of crystal hydrate water appeared at 3545, and 3400 $cm^{-1}$ for gypsum, at 3555 and 3610 $cm^{-1}$ for bassanite (Prasad, 2005). Furthermore, it should be noticed that the peak at 3400 $cm^{-1}$ from $CaSO_4\ 2H_2O$ on the samples of FAS-40, FAS-57, FAS-75, and FAS-87 were much stronger than the peak at 3400 $cm^{-1}$ from condensed water on $CaCO_3$ particles. Therefore it can be inferred that $Ca(NO_3)_2$, $NH_4NO_3$, $CaSO_4\ nH_2O$ (gypsum and bassanite) were produced at 60% RH from the heterogeneous reaction of $NO_2$ with the $CaCO_3$-$(NH_4)_2SO_4$ mixtures.

The spectrum of FAS-0 in Fig. 1d was similar to that in Fig. 1c, while there were considerable changes for spectra of the mixtures as RH increased to 85%. Peaks observed at 981, 998, 1131, 1177 $cm^{-1}$ on the mixtures due to the stretching vibration modes of $SO_4^{2-}$ as well as peaks at 2860, 3064, 3192 $cm^{-1}$ assigned to the stretching vibration modes of $NH_4^+$ indicated the formation of $(NH_4)_2Ca(SO_4)_2\ H_2O$ (koktaite) (Jentzsch et al., 2012). The absorption band of nitrate overlapped with that of koktaite at 749 $cm^{-1}$. It can be inferred that koktaite, an intermediate production of gypsum, was formed rapidly as a result of the interaction of ions in the liquid film after the deliquescence of $(NH_4)_2SO_4$ and surface salts (Cziczo et al., 1997; Lightstone et al., 2000). Additionally, the IR absorption peaks at 1570 $cm^{-1}$ in Figure 1d are much stronger than those at 40% and 60% RH. The positive intensity is likely due to the increasing information of $HCO_3^-$, which is from the decomposition of the bulk $CaCO_3$ under wet conditions. It can be interpreted that the reaction of $NO_2$ can occur not only on the surfaces of $CaCO_3$ and the mixtures but also into the bulk of the samples under wet conditions. Also the acidity of surface condensed water is enhanced as a result of the

formation of $HNO_3$ and the dissolution of $(NH_4)_2SO_4$, which facilitates the decomposition of bulk $CaCO_3$.

In conclusion, $NO_2$ did not show any significant uptake on pure $(NH_4)_2SO_4$ particles (FAS-100) at all the RHs investigated. And the products formed from the heterogeneous reactions of $NO_2$ with $CaCO_3$-$(NH_4)_2SO_4$ mixtures were strongly dependent on RH. $Ca(NO_3)_2$ was produced under both dry and wet conditions, bassanite, gypsum and koktaite were formed depending on RH.

In another set of experiments, the mixture of FAS-57 was exposed to nitrogen without the introduction of $NO_2$ in order to investigate the solid-state reaction of $CaCO_3$ with $(NH_4)_2SO_4$. As shown in Fig. 2, no new absorption bands occurred after exposing to dry nitrogen for 120 min. The weak peak at 1189 cm$^{-1}$ due to $HSO_4^-$ appeared as a main absorption peak and no obvious absorption band due to $CaSO_4 \cdot nH_2O$ could be observed at 40% RH. The results suggested that little reaction occurred between $CaCO_3$ and $(NH_4)_2SO_4$ particles under dry condition and 40% RH. Therefore the chemical interaction of $Ca(NO_3)_2$ with $(NH_4)_2SO_4$ was responsible for the formation of bassanite in these conditions. This is likely due to the fact that $Ca(NO_3)_2$ is more hygroscopic and soluble than $CaCO_3$ particles. Furthermore, absorption bands attributed to bassanite, gypsum, koktaite, and surface water film could be observed at 60% and 85% RH, indicating that a chemical reaction in the coagulation of $CaCO_3$ and $(NH_4)_2SO_4$ particles actually occurred at 60% and 85% RH without the introduction of $NO_2$. This result was in good agreement with the results reported by Mori et al. (1998) that gypsum was formed from the chemical reaction between $(NH_4)_2SO_4$ and $CaCO_3$ with koktaite acting as an intermediate product at 70% RH. In addition, the integrated absorbance of bands between 1100 and 1250 cm$^{-1}$ for the sample of FAS-57 at 60% and 85% RH in Fig. 2 were about fifty percent and seventy percent of those for FAS-57 at corresponding RH in Fig. 1. Thus $CaSO_4 \cdot nH_2O$ and koktaite products could be formed both from the chemical interaction of $(NH_4)_2SO_4$ with $Ca(NO_3)_2$ and the reaction of $(NH_4)_2SO_4$ with $CaCO_3$ at 60% and 85% RH.

**3.2 Uptake coefficients and kinetics**

The formation rates of nitrate on $CaCO_3$ particle surfaces and the mixtures were studied. The nitrate formed during the reaction was presented by the integrated absorbance ($I_A$) of the IR peak area between 1390 and 1250 $cm^{-1}$. The peak at 1043 $cm^{-1}$ was not used to avoid the interruption of the absorptions of sulfates. The integrated nitrate absorbance over the $v_1$ region (1013-1073 $cm^{-1}$) and the $v_3$ region (1250-1390 $cm^{-1}$) could well overlap after the former multiplied by a constant on $CaCO_3$ particle surfaces (Figure S2). Figure 3 represents the integrated absorbance of nitrate as a function of time at different RHs. The formation of nitrate on sample surfaces could be divided into three stages under dry conditions. The integrated absorbance of nitrate increased linearly with time in initial stage and it slowed down at stable stage after a transition period. Furthermore, the lasting time of initial stages for the mixtures decreased nearly linearly with increasing mass fraction of $(NH_4)_2SO_4$ in the mixtures, e.g., it lasted about 80 min for FAS-0 (pure $CaCO_3$ particles), 30 min for FAS-57, 20 min for FAS-75 and 5 min for FAS-93. In another word, the reactive ability of the mixtures in initial stage had a positive linear relation with the $CaCO_3$ content in the mixtures. The possible reasons were that for the reaction of $NO_2$ with $CaCO_3$-$(NH_4)_2SO_4$ mixtures, nitrate was formed by the uptake of $NO_2$ on $CaCO_3$ particle surfaces without the participation of $(NH_4)_2SO_4$ and the reactions limited on the surfaces under dry condition. Moreover, the lasting time of initial stages were extended with increasing RH, e.g., it extended to 80 min for the mixture of FAS-75, to 50 min for the mixture of FAS-93, and even may longer than 120 min for the mixtures with mass fraction of $(NH_4)_2SO_4$ smaller than 57% at 40% RH. The boundaries between initial stages and transition stages became ambiguous at 60% RH and finally disappeared at 85% RH for all the $CaCO_3$-$(NH_4)_2SO_4$ mixtures. This was likely due to the fact that the reaction of $NO_2$ could react into the bulk of the particles under wet conditions.

The integrated absorbance ($I_A$) for nitrate ions on the samples had a linear relationship with the amount of nitrate determined by ion chromatography $\{NO_3^-\}$:

The nitrate ions: $\{NO_3^-\}$ = (integrated absorbance $I_A$) $\times f$ (1)

Here $f$ is conversion factor. It is calculated to be $(2.14\pm0.17)\times10^{17}$ ions/int.abs at 85% RH

and $(3.32\pm0.13)\times10^{17}$ ions/int.abs at 60% RH, 40% RH and dry condition (see Fig. S3). The conversion factor $f$ may change with the chemical environment of surface nitrate which is related to surface condensed water and ion interaction (Li et al., 2010). Then nitrate formation rates d$\{NO_3^-\}$/dt can be calculated from $f$ and the slope of integrated absorbance as a function of time.

As shown in Fig. 4, the initial nitrate formation rates for the samples showed a maximum value under dry condition, whereas the stable formation rates were much slower in this condition. The initial nitrate formation rates increased slightly as RH increased from 40% RH to 60% and 85% RH for the uptake of $NO_2$ on $CaCO_3$ particle surfaces (FAS-0). For the mixtures with mass fraction of $(NH_4)_2SO_4$ larger than 57%, it showed an opposite variation that initial nitrate formation rates at 40% RH were higher than that at 60% RH, followed by that at 85% RH. While for the mixtures with mass fraction of $(NH_4)_2SO_4$ smaller than 43%, the nitrate formation rates increased initially as RH elevated from 40% RH to 60% RH then it decreased obviously as RH increased to 85% RH. The differences in the tendency of nitrate formation rates with RH for the mixtures could be explained by the combined opposite effects from the interaction of $(NH_4)_2SO_4$ with $Ca(NO_3)_2$ or $CaCO_3$ at 60% RH.

Besides, nitrate formation rates decreased more evidently with increasing $(NH_4)_2SO_4$ content at 85% RH and dry condition than at 40% and 60% RH, e.g., the initial nitrate formation rates for the mixture of FAS-93 under dry condition, 40%, 60%, and 85% RH were 47%, 70%, 62%, and 34% of that for FAS-0 at corresponding RH, respectively. Furthermore as RH increased from dry condition to 40% and 60% RH, the initial nitrate formation rates decreased less for the reaction of $NO_2$ with the mixtures than with $CaCO_3$ particles, while it was opposite as RH increased to 85% RH, e.g., the initial nitrate formation rates for FAS-0 at 40%, 60%, and 85% RH were 64%, 67%, and 72% of that under dry condition, respectively, for the mixture of FAS-93, the initial nitrate formation rates at 40%, 60%, and 85% RH were 95%, 87%, and 60% of that under dry condition. In conclusion, the initial nitrate formation rates were accelerated to an extent at 40% and 60% RH, whereas it was inhibited slightly at 85% RH.

The reactive uptake coefficient ($\gamma$) is defined as the rate of the reactive collisions with the surface divided by the total number of surface collisions per unit time (Z).

$$\gamma = \frac{dN(NO_2)/dt}{Z} \tag{2}$$

$$Z = \frac{1}{4} Asurface \left[ NO_2 \right] \sqrt{\frac{8RT}{\pi M_{NO_2}}} \tag{3}$$

Where $N(NO_2)$ is the number of reactive $NO_2$ collisions with the surface, $A_{surface}$ is the effective surface area of samples and $[NO_2]$ is the gas-phase concentration of $NO_2$. R represents the gas constant, T represents the temperature and $M_{NO2}$ is the molecular weight of $NO_2$. The rate of reactive $NO_2$ collision with the surface can be quantified in terms of the nitrate formation rate $d\{NO_3^-\}/dt$, then the reactive uptake coefficients can be calculated by:

$$\gamma = \frac{d\{NO_3^-\}/dt}{Z} \tag{4}$$

The uptake coefficients of $NO_2$ on $CaCO_3$ particles and $CaCO_3$-$(NH_4)_2SO_4$ mixtures were calculated using both BET and geometric surface area, which could be considered as two extreme cases (Ullerstam et al., 2002). The results are listed in Table 2. The initial uptake coefficients corresponding to BET surface area for $NO_2$ on $CaCO_3$ particle surfaces are $(3.34 \pm 0.14) \times 10^{-9}$, $(2.04 \pm 0.07) \times 10^{-9}$, $(2.23 \pm 0.22) \times 10^{-9}$, and $(2.28 \pm 0.17) \times 10^{-9}$ for dry condition, 40%, 60%, and 85% RH, respectively, well consistent with the previous measurement results (Li et al., 2010; Börensen et al., 2000). The $\gamma_{BET}$ is approximately a factor of $10^4$ smaller than the $\gamma_{geometric}$. The $\gamma_{BET}$ for the uptake of $NO_2$ on the mixtures was enhanced with increasing $(NH_4)_2SO_4$ content because of the decrease of BET surface area. On the contrary, the $\gamma_{geometric}$ decreased with increasing $(NH_4)_2SO_4$ content due to the decrease of nitrate formation rates.

The mass concentrations of $NO_3^-$ formed on the samples after reaction with $NO_2$ were detected by IC, as shown in Fig. 5. The $NO_3^-$ mass concentrations for $CaCO_3$ particles are $3.22 \pm 0.17$, $3.31 \pm 0.03$, $3.38 \pm 0.35$, and $3.47 \pm 0.32$ mg/g under dry condition, 40%, 60% and 85% RH, respectively. It suggests that the $NO_3^-$ mass concentration increase slightly with higher RH for the reaction of $NO_2$ with $CaCO_3$ particles. For the $CaCO_3$-$(NH_4)_2SO_4$ mixtures, the

$NO_3^-$ mass concentrations under dry condition are obviously smaller than those at 85% RH, and it exhibits maximum values at 40% or 60% RH. In addition, it should be noticed that the $NO_3^-$ mass concentrations has a negative linear relation with $(NH_4)_2SO_4$ mass fraction in the mixtures under dry condition, the $R^2$ of liner fit is 0.993. This result is in good agreement with the conclusions of Figure 1a and Figure 3 that the reaction of $NO_2$ with $CaCO_3$-$(NH_4)_2SO_4$ mixtures is very similar to the reaction of $NO_2$ with pure $CaCO_3$ particles under dry condition and that $(NH_4)_2SO_4$ has little effects on the formation of $NO_3^-$ in this condition. Moreover, the concentrations of $NO_3^-$ of the mixtures under wet conditions are markedly larger than those under dry condition. The nitrate concentrations for the mixtures of FAS-10 and FAS-20 at 40% and 60% RH are even larger than that for pure $CaCO_3$ particles. The $NO_3^-$ mass concentrations increase much more for the mixtures than for pure $CaCO_3$ particles as RH elevated from dry condition to wet conditions, e.g., the $NO_3^-$ mass concentrations for the mixture of FAS-57 are 3.23±0.09, 3.09±0.14, 2.42±0.07 mg/g at 40%, 60% and 85% RH, respectively, which are increased by a factor of 2.1, 2.0, and 1.6 in comparison with that for FAS-57 under dry condition (1.55±0.08 mg/g). For the reaction of $NO_2$ with FAS-0, the $NO_3^-$ mass concentrations just increase by a factor of 1.03, 1.05, 1.08, as RH increased from dry condition to 40%, 60% and 85% RH, respectively. Besides, no obvious $NO_3^-$ is formed on pure $(NH_4)_2SO_4$ particles under all conditions investigated. These results clearly reveal that the $CaCO_3$-$(NH_4)_2SO_4$ mixtures exhibit promotive effects on nitrate formation in the heterogeneous reaction with $NO_2$ under wet conditions.

The results described above indicate that relative humidity plays a vital role in the heterogeneous reaction of $NO_2$ with $CaCO_3$-$(NH_4)_2SO_4$ mixtures. Under dry condition, little reaction occurs between $CaCO_3$ and $(NH_4)_2SO_4$. Therefore, nitrate formed on the mixtures under dry condition is mainly produced from the reaction of $NO_2$ with $CaCO_3$ particles. At 40% RH, the solid-state reaction between $CaCO_3$ and $(NH_4)_2SO_4$ particles can be neglected, implying that the solid-state reaction has little effects on the heterogeneous reaction. Meanwhile, the chemical interaction of $Ca(NO_3)_2$ with $(NH_4)_2SO_4$ is enhanced with the deliquescence of $Ca(NO_3)_2$, resulting in the formation of microcrystallites of $NH_4NO_3$ and $CaSO_4 \cdot nH_2O$. Consequently, it may help to improve the ionic mobility of the surface ions

(Allen et al., 1996), modify the surface structure and re-expose reactive sites (Al-Hosney and Grassian, 2005). Thus the chemical interaction of $Ca(NO_3)_2$ and $(NH_4)_2SO_4$ particles may exhibits promotive effects on the nitrate formation during the heterogeneous reaction of $NO_2$ with $CaCO_3$-$(NH_4)_2SO_4$ mixtures. The nitrate formation rates and nitrate concentrations increase slightly when RH increased from 40% RH to 60% RH for the mixtures with mass percentage of $(NH_4)_2SO_4$ less than 43%. However, it was opposite for the mixtures with mass percentage of $(NH_4)_2SO_4$ larger than 57%. This could be possibly explained that there is a combined effect of the two opposing effects on nitrate formation from the interaction of $(NH_4)_2SO_4$ with $Ca(NO_3)_2$ or $CaCO_3$ during the heterogeneous reaction of the mixtures with $NO_2$. Since a chemical reaction in the coagulation of $CaCO_3$ with $(NH_4)_2SO_4$ actually occurred without the introduction of $NO_2$ at 60% RH, leading to the formation of $CaSO_4 \cdot nH_2O$. Consequently, $CaCO_3$ particles are partly consumed during the coagulation process and $CaSO_4 \cdot nH_2O$ formed in the coagulation may block reactive sites for further reaction. Thus, the solid state reaction between $CaCO_3$ and $(NH_4)_2SO_4$ particles exhibits inhibiting effects on the formation of nitrate on the mixtures. As for 85% RH, the deliquescence of $(NH_4)_2SO_4$ and surface nitrate leads to more water uptake on the mixture surfaces. The inhibiting effects from the coagulation of $CaCO_3$ and $(NH_4)_2SO_4$ in water film become stronger at 85% RH than at 60% RH, resulting in the decrease of nitrate formation rates and nitrate concentrations at 85% RH in comparison with those at 40% and 60% RH.

**3.3 Mechanism**

According to the experimental observations described above, a reaction mechanism for the heterogeneous reactions of $NO_2$ with $CaCO_3$-$(NH_4)_2SO_4$ mixtures was proposed.

Gas phase $NO_2$ attached to surface OH groups on $CaCO_3$ particle surfaces, as shown in (R1), where (g) is the gas phase and (ads) is the adsorbed phase.

$$\text{S-OH} + NO_2(g) \rightarrow \text{S-OH}{\ldots}NO_2(ads) \tag{R1}$$

Börensen et al. (2000) proposed that two adsorbed-phase $NO_2$ molecules result in surface

nitrate and nitrite products through a disproportionation reaction. Underwood et al. (1999b) suggested that $NO_2$ (g) reacted to form adsorbed nitrite species initially and then react with another surface nitrite or with gas-phase $NO_2$ to form nitrate. Nitrite was detected by FTIR and IC in this study. The reaction process can be described as:

$$2S\text{-}OH\ldots NO_2(ads) \rightarrow S\ldots NO_3^-(ads) + S\ldots NO_2^-(ads) + H_2O \tag{R2}$$

$$2\,S\ldots NO_2^-(ads) \rightarrow S\ldots NO_3^-(ads) + NO(g) \tag{R3}$$

$$S\ldots NO_2^-(ads) + NO_2(g) \rightarrow S\ldots NO_3^-(ads) + NO(g) \tag{R4}$$

Under dry condition, the surface nitrate was in equilibrium with surface adsorbed water and adsorbed $HNO_3$ species (R5). Adsorbed $H_2CO_3$ can exist on $CaCO_3$ particle surfaces (R6) and there was weak chemical interaction between $Ca(NO_3)_2$ and $(NH_4)_2SO_4$ (R7).

$$S\ldots NO_3^-(ads) + S\ldots H_2O(ads) \rightarrow S\ldots HNO_3(ads) + S\text{-}OH \tag{R5}$$

$$2S\ldots HNO_3(ads) + CaCO_3 \rightarrow Ca(NO_3)_2 + S\ldots H_2CO_3(ads) \tag{R6}$$

[revised manuscript text omitted]
}$ $(\times 10^{-9})$ | $\gamma_{geo}$ $(\times 10^{-6})$ | $\gamma_{BET}$ $(\times 10^{-9})$ | $\gamma_{geo}$ $(\times 10^{-6})$ | $\gamma_{BET}$ $(\times 10^{-9})$ | $\gamma_{geo}$ $(\times 10^{-6})$ | $\gamma_{BET}$ $(\times 10^{-9})$ | $\gamma_{geo}$ $(\times 10^{-6})$ |
| 0 | 3.34±0.14 | 10.4±0.44 | 2.04±0.07 | 6.36±0.22 | 2.23±0.22 | 6.94±0.69 | 2.28±0.17 | 7.10±0.53 |
| 10 | 3.19±0.21 | 9.83±0.65 | 2.06±0.21 | 6.34±0.45 | 2.25±0.14 | 6.91±0.43 | 2.13±0.41 | 6.56±1.26 |
| 20 | 3.77±0.24 | 9.54±0.61 | 2.51±0.34 | 6.28±0.86 | 2.74±0.42 | 6.87±1.06 | 2.00±0.21 | 5.63±0.53 |
| 40 | 5.34±0.17 | 9.25±0.29 | 3.50±0.42 | 6.07±0.72 | 3.67±0.48 | 6.36±0.83 | 3.15±0.28 | 5.46±0.49 |
| 57 | 6.82±0.33 | 8.38±0.41 | 4.70±0.51 | 5.78±0.63 | 4.47±0.26 | 5.49±0.32 | 4.15±0.53 | 5.10±0.65 |
| 75 | 7.74±0.94 | 6.94±0.84 | 6.12±0.37 | 5.49±0.23 | 5.80±0.53 | 5.20±0.48 | 4.26±0.31 | 3.82±0.28 |
| 87 | 9.04±0.73 | 5.78±0.46 | 7.68±0.50 | 4.92±0.32 | 7.22±0.63 | 4.63±0.40 | 4.83±0.46 | 3.10±0.19 |
| 93 | 14.4±1.07 | 4.90±0.36 | 13.6±0.93 | 4.63±0.32 | 12.7±0.81 | 4.34±0.28 | 7.48±0.82 | 2.55±0.28 |

[Figure]

**Figure 1.** DRIFTS spectra of CaCO$_3$ particles (FAS-0), CaCO$_3$-(NH$_4$)$_2$SO$_4$ mixtures (FAS-10 - FAS-93), and (NH$_4$)$_2$SO$_4$ particles (FAS-100) after reaction with NO$_2$ at (a) dry condition, (b) 40% RH, (c) 60% RH, (d) 85% RH for 120 min. The concentration of NO$_2$ was $2.6 \times 10^{15}$ molecule cm$^{-3}$.

[Figure]

**Figure 2.** In situ DRIFTS spectra of surface products when the mixture of FAS-57 were exposed to nitrogen at dry condition (black), 40% RH (green), 60% RH (blue) and 85% RH (red) for 120 min.

[Figure]

**Figure 3.** The integrated absorbance of the peak area between 1390 and 1250 cm$^{-1}$ for nitrate on pure CaCO$_3$ particle surfaces (FAS-0), and CaCO$_3$-(NH$_4$)$_2$SO$_4$ mixtures (FAS-10 - FAS-93) at (a) dry condition, (b) 40% RH, (c) 60% RH, and (d) 85% RH. The NO$_2$ concentration was $2.6 \times 10^{15}$ molecule cm$^{-3}$.

[Figure]

**Figure 4**. Initial nitrate formation rates at dry condition (rhombus), 40% RH (triangle), 60% RH (fall triangle), 85% RH (roundness) and stable nitrate formation rates (pentagon) under dry condition versus the mass percentage of $(NH_4)_2SO_4$ in the mixtures. The data points and the error bars are the average value and the standard deviation of three duplicate experiments.

[Figure]

**Figure 5**. The mass concentration of $NO_3^-$ for $CaCO_3$ particles and the $CaCO_3$-$(NH_4)_2SO_4$ mixtures after reacted with $NO_2$ for 120 min as a function of the mass percentage of $(NH_4)_2SO_4$ in the mixtures. The data points and the error bars are the average value and the standard deviation of three duplicate experiments.

[Figure]

**Figure 6.** Schematic illustrating the possible heterogeneous processes of $NO_2$ with $CaCO_3$-$(NH_4)_2SO_4$ mixtures and the possible atmospheric implications.